# Daily humidity oscillation regulates the circadian clock to influence plant physiology

Musoki Mwimba[1,2], Sargis Karapetyan[2,3], Lijing Liu[1,2], Jorge Marqués[1,2], Erin M. McGinnis[1,2], Nicolas E. Buchler[2,3,4] & Xinnian Dong[1,2]

Early circadian studies in plants by de Mairan and de Candolle alluded to a regulation of circadian clocks by humidity. However, this regulation has not been described in detail, nor has its influence on physiology been demonstrated. Here we report that, under constant light, circadian humidity oscillation can entrain the plant circadian clock to a period of 24 h probably through the induction of clock genes such as *CIRCADIAN CLOCK ASSOCIATED 1*. Under simulated natural light and humidity cycles, humidity oscillation increases the amplitude of the circadian clock and further improves plant fitness-related traits. In addition, humidity oscillation enhances effector-triggered immunity at night possibly to counter increased pathogen virulence under high humidity. These results indicate that the humidity oscillation regulates specific circadian outputs besides those co-regulated with the light-dark cycle.

[1] Howard Hughes Medical Institute, Duke University, Durham, NC 27708, USA. [2] Department of Biology, Duke University, PO Box 90338, Durham, NC 27708, USA. [3] Department of Physics, Duke University, Durham, NC 27708, USA. [4] Department of Molecular Biomedical Sciences, North Carolina State University, Raleigh, NC 27606, USA. Correspondence and requests for materials should be addressed to X.D. (email: xdong@duke.edu)

Life on earth is fundamentally dictated by the daily and yearly rhythms produced as a result of Earth's rotation and revolution around the Sun. Living organisms perceive these rhythms and use biological circadian clocks to help tell time and schedule physiological processes in anticipation of regular changes in the environment[1–4]. In plants, the circadian clock is a complex network of transcriptional and translational feedback loops[5,6]. At the core of the plant circadian clock are two partially redundant morning-phased transcription factors (TFs)[7], CIRCADIAN CLOCK ASSOCIATED 1 (CCA1) and LATE ELONGATED HYPOCOTYL (LHY), which repress their own expression as well as the expression of their repressors, the PSEUDO-RESPONSE REGULATORS (PRR1, PRR5, PRR7 and PRR9)[8]. PRR1, better known as TIMING OF CAB EXPRESSION 1 (TOC1), is an evening-phased TF, which, together with CCA1 and LHY, forms the core feedback loop[8,9]. This CCA1/LHY-TOC1 core loop is further connected with the morning loop (i.e., PRR5, PRR7 and PRR9) and the evening loop (i.e., PRR3, EARLY FLOWERING 3, EARLY FLOWERING 4, LUX ARRHYTHMO and GIGANTEA) to create the complex architecture of the plant circadian clock[8].

Early investigations had alluded to effects of light, temperature and air humidity on circadian clocks[10,11]. Yet, most studies of plant and animal circadian clocks have been conducted using only light or temperature as a clock entraining signal even though correlations between humidity changes and potential clock outputs, including metabolism[12–15], animal behaviours (e.g., the biting activity of mosquitos[16–19]) and incidence of plant diseases[20,21] have been reported. The extent to which daily humidity oscillation directly regulates the circadian clock has not been investigated. Our study reveals that in the absence of light and temperature cues, air humidity entrains the plant circadian clock probably through high humidity-mediated induction of *CCA1*; whereas, in the presence of light-dark cycles, oscillating air humidity increases the amplitude of clock genes, further improves plant fitness-related traits and enhances resistance against evening-inoculated pathogens. Therefore, this study not only demonstrates humidity oscillation as a regulator of the circadian clock but also shows its specific impacts on plant physiology.

## Results

**Humidity oscillation regulates physiological clock outputs**. To test whether humidity is a regulator of the plant circadian clock, we first examined the influence of daily humidity oscillation on plant physiology. We kept the temperature constant at 22 °C so that relative humidity (RH) could be used as a proxy for vapour pressure deficit (VPD), which is known to regulate plant transpiration[22–24]. Additionally, we performed our experiments in constant light (LL) to exclude the influence of the light-dark cycle. We first compared plants grown under 22 °C in either constant RH of 50% or 90% to plants grown under 22 °C in 12 h 50%/12 h 90% oscillating RH. The oscillating RH was designed to simulate weather conditions observed in Harvard Forest (latitude: +42.53311, longitude: −72.18968) during the equinoxes, which had a 24-h period and 40% RH range (Supplementary Fig. 1a–c). To minimize confounding results due to temperature changes, we grew plants in humidity- and temperature-controlled Percival chambers (Supplementary Fig. 2a,b; Supplementary Table 1). We found the biggest increases in leaf expansion and biomass in plants grown under 12 h 50%/12 h 90% oscillating RH (Fig. 1a, b), suggesting that beyond the known effects of high humidity on plants, such as the opening of stomata[12,25] and increasing $CO_2$ assimilation[26,27], humidity oscillation has additional influences on plant physiology. To test whether these influences could be circadian clock-regulated, we compared wild-type (WT) plants to the clock core loop triple mutant *cca1 lhy toc1*. Plants were grown in LL at 22 °C under either 12 h 50%/12 h 90% oscillating RH or 70% constant RH, so that they were exposed to the same total amount of humidity. We found that WT plants accumulated more biomass and flowered earlier under oscillating humidity than under 70% RH and that these effects were dependent on core clock genes (Fig. 1c, d). These results suggest that the daily oscillation of air humidity could, in principle, be a regulator of the plant circadian clock.

**Humidity oscillation entrains the circadian clock in LL**. Next, we investigated whether humidity could entrain the circadian clock, i.e. synchronize the clock in the absence of other environmental signals. We examined synchronicity of clock gene expression in transgenic *Arabidopsis* plants carrying luciferase

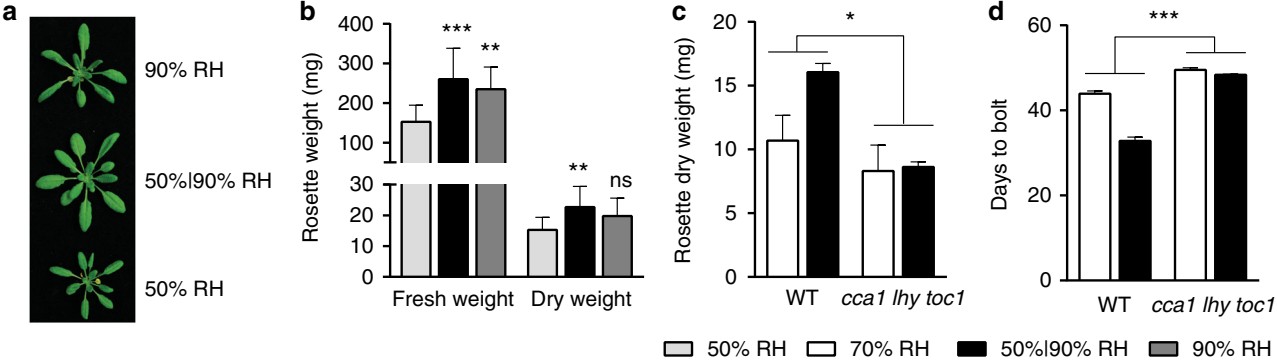

**Fig. 1** Humidity oscillation influences plant physiology in a core clock gene-dependent manner under constant light conditions. **a**, **b** Photograph (**a**) and weight (**b**) of leaf-rosette of 3-week-old plants grown under constant light (LL) with constant 50% relative humidity (RH), 90% RH, or 12 h 50%/12 h 90% oscillating RH. Data are shown as mean ± s.d. of $n \geq 11$ plants; one-way ANOVA followed by Bonferroni's multiple comparisons correction, **$p < 0.01$, ***$p < 0.001$; ns not significant. This experiment was repeated two times with similar results. **c** Dry weight of leaf-rosette of 3-week-old WT and the core clock gene mutant *cca1 lhy toc1* plants grown under LL with constant 70% RH or 12 h 50%/12 h 90% oscillating RH. Data are shown as mean ± s.d. of three experimental replicates representing at least 45 plants from a single parent line. Data were combined using linear mixed effect model (lme4) with experiment as random effects; two-way ANOVA, *$p < 0.05$. **d** Flowering time of plants grown under LL with constant 70% RH or 12 h 50%/12 h 90% oscillating RH. Data are shown as mean ± s.d. of three experimental replicates representing at least 78 plants from a single parent line. Data were combined using linear mixed effect model (lme4) with experiment as random effects; two-way ANOVA, ***$p < 0.001$

reporters driven by promoters of the clock genes *CCA1* (*CCA1p:LUC*), *LHY* (*LHYp:LUC*) and *TOC1* (*TOC1p:LUC*). In order to distinguish the influence of humidity oscillations from those of temperature and light cycles on the circadian clock, we first grew plants in the humidity- and temperature-controlled Percival chambers (Supplementary Fig. 2a,b) and then imaged them in a custom-built chamber containing two compartments in which we could independently oscillate humidity while keeping temperature within a range of 1 °C (Supplementary Fig. 2c), which is well below the lowest temperature range (4 °C) reported to entrain the *Arabidopsis* circadian clock[28].

From seed germination, we grew plants in two Percival growth chambers set to LL 22 °C conditions with humidity kept at constant 50% RH in one chamber and oscillating (12 h 50%/12 h 90% RH) in the other chamber. After 3 weeks, plants were moved to the imaging chamber and clock gene expression was measured by luciferase reporter activities in the area around the shoot apical meristem, which displays robust and synchronous waveforms[29]. We found persisting *CCA1p:LUC*, *LHYp:LUC* and *TOC1p:LUC* rhythms in plants entrained with oscillating humidity even after they were released into free-running conditions (LL 22 °C 50% RH) (Fig. 2a). A larger fraction of plants pre-grown under oscillating humidity were rhythmic and their phases were more synchronized when compared to those of plants pre-grown under constant humidity (Fig. 2b; Supplementary Table 2), which indicates entrainment. This entrainment was verified by quantitative reverse transcription-PCR to rule out possible influence of humidity on the luciferase reporter activity (Supplementary Fig. 3a–d). In addition to the core circadian clock genes, the humidity oscillation also entrained clock output genes, such as the morning-phased *CATALASE 2* (*CAT2*) and the evening-phased *COLD*, *CIRCADIAN RHYTHM* and *RNA BINDING 2* (*CCR2*) and *CATALASE 3* (*CAT3*) (Fig. 2c, d; Supplementary Table 2; Supplementary Fig. 3e, f), suggesting that entrainment by humidity oscillation influences clock functions.

In the entrainment experiments shown in Fig. 2a–d and Supplementary Fig. 3, we were intrigued by the phases of the clock and clock output genes, which were at their respective antiphases, i.e., morning-phased genes *CCA1*, *LHY* and *CAT2* peaked at the low-to-high humidity transition, which corresponded to dusk in nature, whereas the opposite occurred to the evening-phased genes *TOC1*, *CAT3* and *CCR2*. To confirm that humidity oscillation could set the phase of the clock, we grew two populations of plants from germination at 22 °C and 50% RH but under antiphasic light regimens (LD or DL). After 2 weeks, plants were temporarily moved to free-running conditions (LL and 50% RH) at their respective dawn for imaging and then moved to LL 12 h 50%/12 h 90% RH humidity oscillation conditions, which would have corresponded to LD in nature (Fig. 2e). We found that the humidity oscillation shifted the phase of plants initially pre-grown under LD to their antiphase whereas plants initially pre-grown under DL did not experience a phase-shift (Fig. 2f), supporting the hypothesis that humidity oscillation sets the phase of the circadian clock under LL.

To gain insight into the mechanism by which humidity sets the phase of the clock, we investigated whether humidity could perturb the endogenous rhythm of the core clock genes. *Arabidopsis* plants carrying luciferase reporters were grown for 3 weeks under 12 h light/12 h dark (LD) and 50% RH, and then released to DD and 50% RH. After 54 h under DD, i.e., 6 h after the subjective dawn when humidity is unlikely to be high in nature, we exposed these plants to a treatment of 2-h 90% RH and detected an immediate induction of the morning-phased *CCA1p:LUC* reporter (Fig. 2g, h) as well as the *CCA1* mRNA (Fig. 2i). Interestingly, the other morning core clock gene *LHY* was insensitive to the 2-h pulse of high humidity treatment

(Fig. 2g–i) even though it could be entrained by repeated 12 h 50%/12 h 90% RH cycles (Fig. 2a, b, f; Supplementary Fig. 3). These results suggest that high humidity could set the phase of the clock through the induction of *CCA1*.

**Influence of oscillating humidity on the *prr7 prr9* mutant**. To investigate whether oscillating humidity potentially entrains the clock via changes in temperature, a known entraining signal, we tested the *prr7 prr9* double mutant, which is unresponsive to 12 °C/22 °C temperature entrainment[30]. We subjected the double mutant to humidity oscillation (12 h 50%/12 h 90% RH) and, as a control, to air temperature oscillation (12 h 21 °C/12 h 22.5 °C), which covers the range of ambient temperature fluctuations observed during humidity oscillation in our chambers (Supplementary Fig. 2). We found that while both humidity and the 1.5 °C air temperature cycles could entrain *LHYp:LUC* in WT plants, only humidity oscillation could entrain the clock reporter in *prr7 prr9* (Supplementary Fig. 4a,b; Supplementary Table 2).

Light, air temperature and humidity are also known to influence leaf temperature through radiation, heat exchange and transpiration, respectively. Therefore, it is possible for their perception to partially converge on leaf temperature. To test this hypothesis, we subjected the *prr7 prr9* mutant to humidity oscillation and air temperature oscillation, which produced similar oscillations in leaf temperature. Using a FLIR A320sc infrared (IR) camera, we determined the relationship between air temperature/humidity and leaf temperature in our growth chamber (Supplementary Fig. 4c). We found that the range of leaf temperature oscillation under 21 °C and 50%/90% RH corresponded to the one under 21 °C/22.5 °C and 50% RH (Supplementary Fig. 4d). Yet, we observed entrainment of *prr7 prr9* under 21 °C with 12 h 50%/12 h 90% RH, but not under 50% RH with 12 h 21 °C/12 h 22.5 °C (Supplementary Fig. 4b,e).

**Humidity increases the amplitude of core clock genes in LD**. Since light and humidity cycles coexist in nature, we hypothesized that similar to the synergistic effect by simulated light and temperature cycles on the *Drosophila*'s clock[31], humidity oscillation may increase the amplitudes of *CCA1*, *LHY* and *TOC1*, and improve fitness-related traits in LD. To test this hypothesis, we compared *CCA1p:LUC*, *LHYp:LUC* and *TOC1p:LUC* expression in plants grown under LD with either constant 50% RH or 12 h 50% RH/12 h 90% RH cycle with the 2-h overlaps observed in nature (i.e., 2-h 90% RH with light at dawn and 2-h 50% RH with darkness at dusk; Supplementary Fig. 1d). For the remainder of the manuscript, we term this combined humidity and light cycles as "simulated natural conditions". We found that the simulated natural conditions increased the expression amplitudes of core clock genes as determined by the clock-driven reporter activities as well as the endogenous clock gene mRNA levels (Fig. 3a–c, Supplementary Fig. 5). Interestingly, the 2-h overlap between light and humidity cycles appears to be required for the interplay between light and humidity on the circadian clock, because when the humidity cycles were set in complete antiphase with the light cycles, little amplitude increase was observed (Fig. 3d–f). Although the humidity-sensing mechanism has yet to be elucidated, these data suggest that humidity may increase the amplitude of the light-driven clock by targeting some morning clock genes such as *CCA1*.

**Simulated natural conditions improve fitness-related traits**. Next, we investigated the influence of simulated natural conditions on several physiological traits previously shown to be affected by the clock[32–35]. We compared biomass accumulation, flowering time and seed production in plants grown under light-dark cycles with either simulated natural conditions or

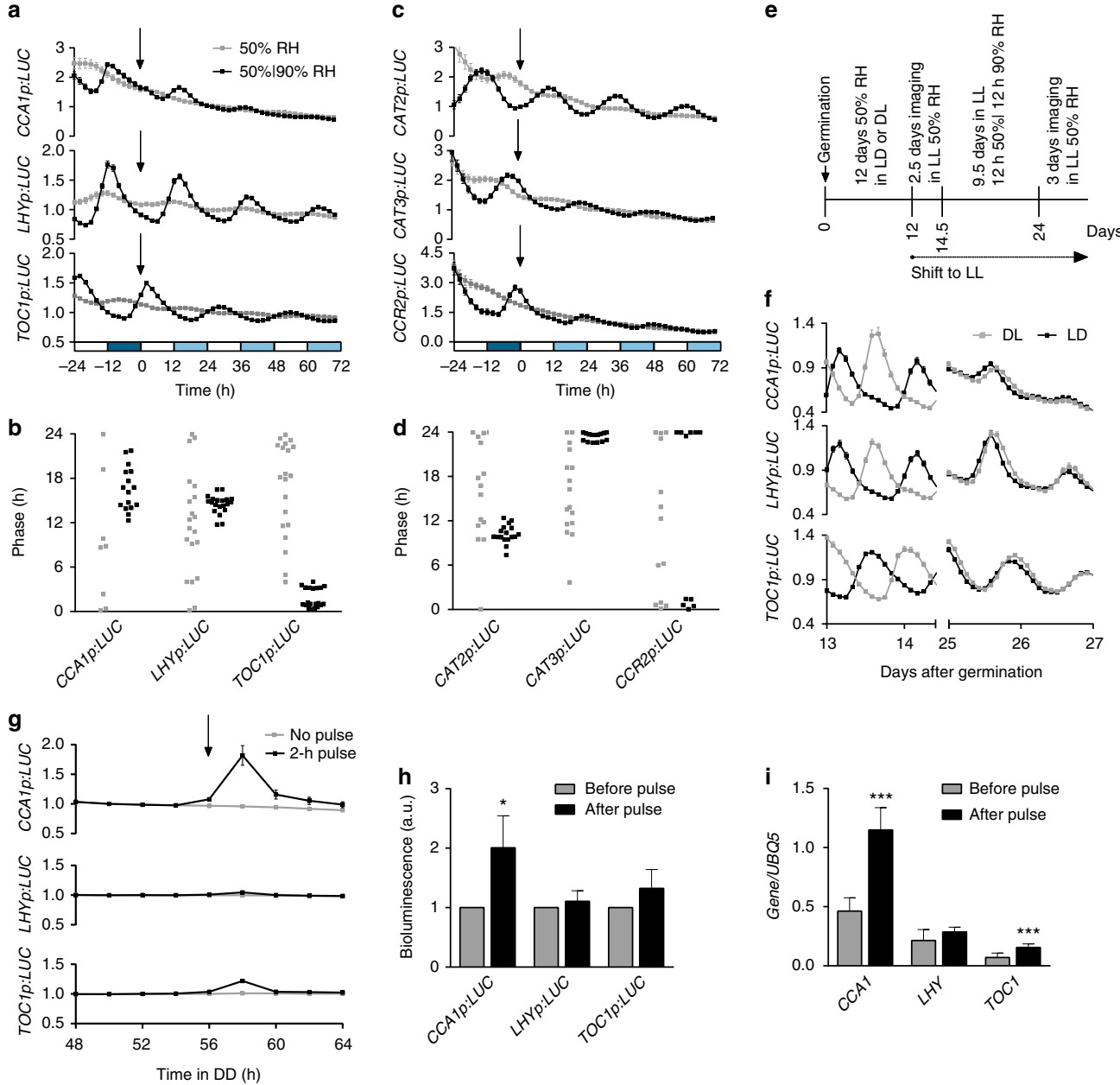

**Fig. 2** Humidity oscillation entrains the circadian clock in LL. **a**–**d** Luciferase reporter activity rhythms (**a**, **c**) and calculated circadian phases (**b**, **d**) of core clock genes, *CCA1*, *LHY* and *TOC1* (**a**), and circadian clock output genes, *CAT2*, *CAT3* and *CCR2* (**c**), in WT plants grown under LL at 22 °C with either constant 50% RH (grey) or 12 h 50%/12 h 90% RH (black). Luciferase activity was normalized to the peak value between time −26 and −4 h. Circadian phases were calculated by performing FFT-NLLS[62] to the data between 0 and 72 h with the period constrained between 18 and 30 h. White bars, 50% RH; dark blue bars, 90% RH; light blue bars, subjective 90% RH; black arrow, transition from entrainment conditions to free-running conditions. Data are shown as mean ± s.e.m.; $n \geq 20$ plants. Relative amplitude error values of ≤0.7 were considered rhythmic. These experiments were repeated three times with similar results. See Supplementary Fig. 3. **e** Workflow describing the experiment of phase setting by humidity. **f** Luciferase reporter activity rhythms of *CCA1*, *LHY* and *TOC1* in response to conditions described in **e**. Plants were grown under DL (grey) or LD (black). Data are shown as mean ± s.e.m.; $n \geq 12$ plants. **g** Luciferase reporter activity rhythms of *CCA1p:LUC*, *LHYp:LUC* and *TOC1p:LUC* in response to a 2-h pulse of 90% RH at 54 h in DD. Luciferase reporter activity was normalized to the average value between 48 and 52 h. DD constant dark; black arrow, start of the 2-h 90% RH pulse. Data are shown as mean ± s.e.m.; $n \geq 8$ plants. The experiment was repeated three times with similar results. **h**, **i** Luciferase activities (**h**) and endogenous clock gene expression (**i**) in response to a 2-h pulse of 90% RH at 54 h in DD. Data are shown as mean ± s.d.; *t*-test, $*p < 0.05$, $***p < 0.001$

constant 70% RH. We found that simulated natural conditions increased plant biomass accumulation and shortened flowering time in WT (Fig. 4a, b). More importantly, despite having no detectable influence on seed quantity, humidity oscillation improved seed quality as determined by seed mass[36] (Fig. 4c, d). These influences were not observed in the *cca1 lhy toc1* triple mutant (Fig. 4a, b, d), which is known to have near WT growth

phenotype in LD[37]. These results indicate that simulated natural conditions influence plant physiology in a core clock gene-dependent manner.

**Simulated natural conditions influence plant immunity.** High humidity is known to correlate with disease outbreaks in

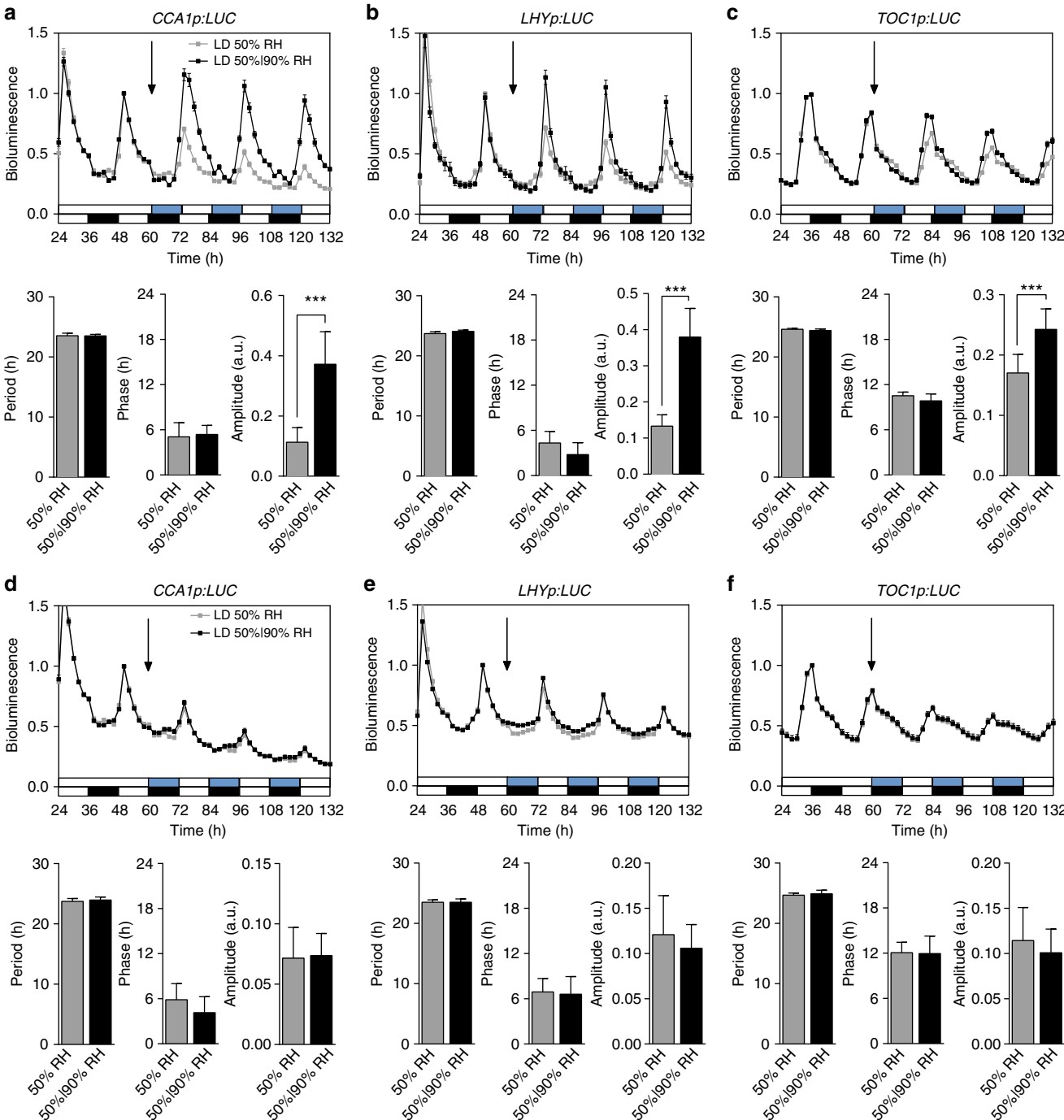

**Fig. 3** Simulated natural conditions increase the amplitude of the circadian clock under the light-dark cycle. **a–c** *CCA1p:LUC* (**a**), *LHYp:LUC* (**b**) and *TOC1p: LUC* (**c**) activity rhythms (top) and calculated periods, circadian phases and amplitudes (bottom) in WT grown under LD with constant 50% RH (grey) or simulated natural conditions (black). Simulated natural conditions represent 12 h 50% RH/12 h 90% RH cycles with the 2-h overlap of high humidity and light in the morning as observed in nature. Luciferase activity was normalized to the peak value between 40 and 60 h. Periods, circadian phases and amplitudes were calculated by FFT-NLLS analysis of the data between 60 and 132 h. White and black bars represent light and dark, respectively; white and blue bars represent 50% RH and 90% RH, respectively; black arrow, start of humidity entrainment conditions. Data are shown as mean ± s.e.m. and mean ± s.d. for activity rhythms and their calculated parameters, respectively; *n* = 24 plants; *t*-test, ***$p < 0.001$. Relative amplitude error values of ≤0.7 were considered rhythmic. This experiment was repeated three times with similar results. **d–f** *CCA1p:LUC* (**d**), *LHYp:LUC* (**e**) and *TOC1p:LUC* (**f**) activity rhythms (top) and calculated periods, circadian phases and amplitudes (bottom) in WT grown under LD with constant 50% RH (grey) or 12 h 50%/12 h 90% RH (blue) with no overlap of light and high humidity in the morning. Data are shown as mean ± s.e.m. and mean ± s.d. for activity rhythms and their calculated parameters, respectively; *n* = 24 plants. Relative amplitude error values of ≤0.7 were considered rhythmic. This experiment was repeated three times with similar results

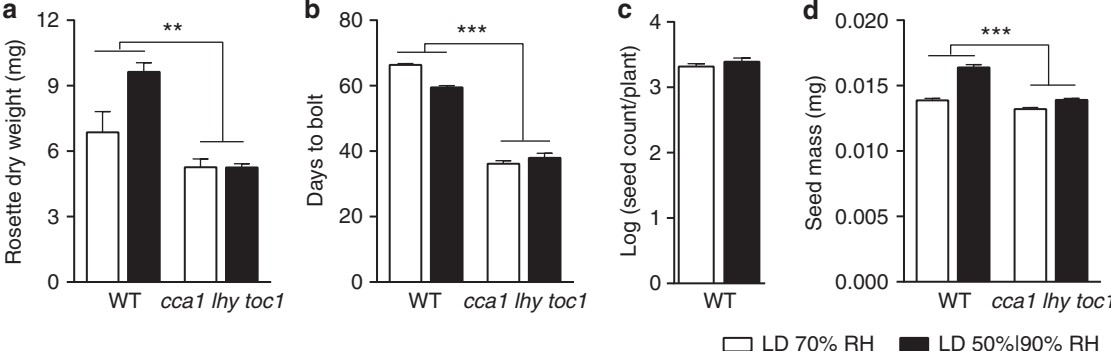

**Fig. 4** Simulated natural conditions improve plant fitness-related traits. Leaf-rosette dry weight (**a**), flowering time (**b**), seed count (**c**) and seed mass (**d**) of plants grown under LD with either constant 70% RH (white) or simulated natural conditions (black). Data are shown as mean ± s.d. of three experimental replicates representing at least 45 plants (**a**), 75 plants (**b**), 36 plants (**c**) and 70 plants (**d**) from a single parent line. Data were combined using linear mixed effect model (lme4) with experiment as random effects; two-way ANOVA, **$p < 0.01$, ***$p < 0.001$

plants[20,38], and the outcome of an infection is determined not only by the robustness of plant immunity but also by the virulence of the pathogen[39,40]. We hypothesized that humidity might regulate the virulence of pathogens. To test this hypothesis, we examined whether high humidity could induce the expression of *hrpL*, the sigma factor gene required for the transcription of the type III secretion system (T3SS) genes involved in *Pseudomonas syringae* (*Ps*) delivery of effectors into the plant cell[40,41]. We subjected *Ps* pv. *tomato* DC3000 (*Pst*) colonies to a 6-h treatment of 95% RH and found that high humidity induced the expression of *hrpL* in bacteria grown on plates made with either the King's B medium or the hrp minimal medium (HMM), which is known to induce *hrpL* expression through its reduced carbon content and low pH[42,43] (Supplementary Fig. 6a). Moreover, high humidity also induced the expression of bacterial virulence effector genes *hopD1* and *hopN1*, but not of the housekeeping gene *gap1*[39], supporting our hypothesis that high humidity could induce pathogen virulence. The increase in *hrpL* expression in HMM suggests that high humidity may activate the bacterial T3SS gene expression through a mechanism other than that involving medium composition and pH.

The humidity-mediated induction of bacterial T3SS genes also led to the hypothesis that humidity oscillation may affect the regulation of the plant immunity by the circadian clock at the level of effector-triggered immunity (ETI), which is elicited by the recognition of pathogen effectors delivered inside the plant cell. To further test whether an influence of humidity oscillation on plant immunity is ETI-specific, we also assessed pattern-triggered immunity (PTI), which is elicited by the recognition of microbe-associated molecular patterns on the host cell surface. We pressure-infiltrated the pathogen into *Arabidopsis* leaves to bypass stomatal regulation on defence[44]. Infection was performed at 1 h after dawn and 1 h after dusk to capture the sensitivity of plant immunity to the time of day (Fig. 5a). We found that the simulated natural conditions did not influence PTI in leaves infiltrated with *Pst* or the T3SS mutant, *Pst hrpL*(−), which is deficient in effector delivery (Supplementary Fig. 6b). However, ETI elicited by *Pst* carrying the AvrRpt2 effector (*Pst AvrRpt2*) was stronger under oscillating humidity than under constant humidity in the evening-inoculated leaves (Fig. 5b). Moreover, the ETI-associated programmed cell death (PCD) of the infected cells was also more pronounced under simulated natural conditions in the evening-inoculated leaves (Fig. 5c). These differences were not observed in the morning-inoculated leaves (Fig. 5b, c), suggesting a time-of-the-day sensitivity to ETI induction under the simulated natural conditions. These effects by oscillating humidity were through the canonical ETI pathway

because ETI in the AvrRpt2 immune receptor mutant, *rps2*, was insensitive to the simulated natural conditions (Fig. 5b, c). We have also obtained similar results with ETI mediated by the immune receptor RRS1 in response to the effector AvrRps4 (Supplementary Fig. 6c), indicating that the time-of-the-day sensitivity to ETI induction is not restricted to the RPS2 receptor.

Finally, we subjected plants pre-grown under 50% RH or simulated natural conditions to 50% RH, 90% RH or simulated natural conditions after being challenged by pathogens at 1 h after dusk (Fig. 5d). We found that similar to the report by Xin et al.[45], exposure to constant 90% RH post inoculation suppressed PCD (Fig. 5e; Supplementary Fig. 6d), indicating that the oscillation of humidity, not high humidity, is required for enhancing ETI at night. Moreover, we found that exposure to simulated natural conditions, either before or after pathogen inoculation, could increase PCD (Fig. 5e; Supplementary Fig. 6d). The enhanced ETI in plants moved to constant 50% RH after being pre-grown under simulated natural conditions implies that plants anticipate humidity oscillation in LD and suggests that the regulation of ETI by humidity oscillation is clock regulated.

## Discussion

In this study, we show that the daily oscillation of air humidity regulates the plant circadian clock and influences plant physiology in a core clock gene-dependent manner. In the absence of the light cycle, humidity oscillation entrains the circadian clock to its antiphase (Fig. 2, Supplementary Figs. 3 and 4), whereas under simulated natural conditions, it increases the amplitude of the clock and further improves fitness-related traits (Figs. 3 and 4). Our entrainment results support the existence of a humidity-sensitive oscillator within the circadian clock, but with *CCA1* and *LHY*, the two parallel morning genes, responding to humidity differently; *CCA1* responded rapidly to humidity, whereas *LHY* did not (Fig. 2g). Rhythmic expression of *LHY* and *TOC1* seen in the oscillating humidity could conceivably be driven by the altered expression of *CCA1*. This suggests that the humidity regulation of the circadian clock might be distinct from the regulation by light and temperature, which both involve phytochromes as sensors and have similar effects on *CCA1* and *LHY*[46–49]. In addition, we detected a novel effect on ETI (Fig. 5, Supplementary Fig. 6), indicating that the ability to sense oscillating humidity might allow the clock to influence distinct physiological processes that are not controlled by the light/temperature-entrained clock.

It is important to determine how plants perceive humidity oscillation and how this signal is transduced to the clock. Since it is likely that the humidity perception involves leaf transpiration

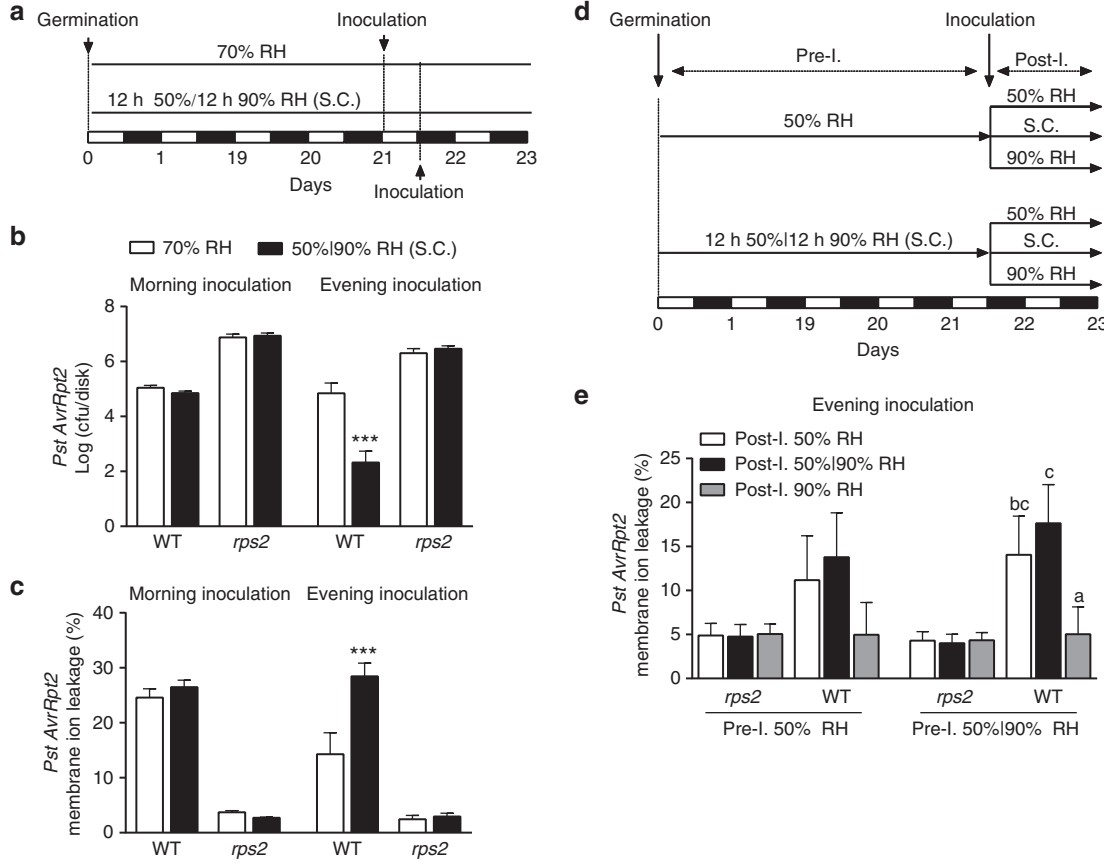

**Fig. 5** Simulated natural conditions improve ETI in evening-inoculated leaves. **a** Workflow describing morning and evening infection experiments in **b** and **c** (Supplementary Fig. 6b, c). S.C. simulated natural conditions. **b** Quantification of *Pst AvrRpt2* growth at 3 days post inoculation in 3-week-old WT and *rps2* (AvrRpt2 receptor mutant) plants grown under LD under 70% RH (grey) or simulated natural conditions (black). Leaves were infiltrated with bacteria at $OD_{600nm} = 0.001$ in the morning (ZT 1 h) or in the evening (ZT 13 h). cfu, colony-forming unit, was determined at 3 dpi. Data are shown as mean ± s.d. of three experimental replicates representing at least 24 plants from a single parent line. Data were combined using linear mixed effect model (lme4) with experiment as random effects; two-way ANOVA; ***$p < 0.001$. **c** Ion leakage measurement performed at 36 h post *Pst AvrRpt2* inoculation of 3-week-old plants grown under LD under 70% RH (grey) or simulated natural conditions (black). Leaves were infiltrated with bacteria at $OD_{600nm} = 0.001$ in the morning (ZT 1 h) or in the evening (ZT 13 h). Data are shown as mean ± s.d. of three experimental replicates representing at least 18 plants from a single parent line. Data were combined using linear mixed effect model (lme4) with experiment as random effects; two-way ANOVA, ***$p < 0.001$. **d** Workflow describing the evening infection experiment in **e** and Supplementary Fig. 6d. **e** Ion leakage measurement performed at 36 h post *Pst AvrRpt2* inoculation of 3-week-old plants grown under LD with 70% RH (grey) or simulated natural conditions (black). Leaves were infiltrated with bacteria at $OD_{600nm} = 0.001$ in the evening (ZT 13 h). Data are shown as mean ± s.d. of three experimental replicates representing at least 18 plants from a single parent line. Data were combined using linear mixed effect model (lme4) with experiment as random effects; *t*-test, the same letters above bars denote values that were statistically similar

due to changes in VPD, we investigated whether humidity entrainment was dependent on leaf temperature, a consequence of leaf transpiration. Temperature is a known entraining signal. If a correlated oscillation in leaf temperature caused by the known effect of humidity on transpiration was sufficient to entrain the clock, it would be expected to cause rhythms in the same phase as observed in our experiments and the *prr7 prr9* double mutant, which is unresponsive to changes in air temperature, may also be unresponsive to changes in humidity. However, our data showed that humidity is still able to entrain rhythms in the temperature-insensitive *prr7 prr9* mutant (Supplementary Fig. 4). We, therefore, suggest that humidity may act via a temperature-independent mechanism. However, a pleiotropic effect of the mutant background on the clock cannot be completely ruled out. Even if humidity does not regulate the clock through leaf temperature, it is still possible that the humidity perception involves other effects of leaf transpiration or sensors shared with the temperature signalling similar to what was recently reported in Drosophilae[50,51]. It also remains unknown how, under simulated natural conditions, oscillating humidity increases the amplitude of the light-driven clock without perturbing its phase. Elucidating this mechanism may require identification of all the humidity targets in the clock, besides *CCA1*, and mathematical modelling of the clock signalling network. Such information may also shed light on the question whether humidity is a novel environmental zeitgeber distinct from light and temperature.

Our study supports previous findings that the circadian clock not only allows plants to anticipate infection at the time when it is most likely to occur[44,52,53] but also gates the immune response when infection does occur to minimize conflicts with growth and development[54]. Previously, we found that untimely induction of defence by the immune signal salicylic acid (SA) at night could lead to a severe loss of plant fresh weight. This was likely due to SA-mediated repression of genes in water transport, such as 9 of the 26 *Arabidopsis* aquaporin genes[54]. Water availability is not only essential for plant growth but also critical for pathogen virulence as recently demonstrated for *Pst* whose proliferation in the apoplast requires both establishment of aqueous conditions

by its effectors and high humidity in the environment[45]. Therefore, humidity may function as a cue to allow scheduling of defence and growth activities with minimal fitness cost.

The daily oscillation of humidity is a universal natural phenomenon experienced by all terrestrial organisms. A thorough investigation of humidity as a regulator of circadian clock in other species may further advance the field of chronobiology by uncovering new clock components, or even organisms to which humidity might be a primary zeitgeber. In addition to our finding that high humidity can activate T3SS in bacteria (Supplementary Fig. 6a), air humidity is known to greatly influence organisms such as insects with high surface to body volume ratios, which make them vulnerable to rapid desiccation[55]. This is in line with the early observations that insect activity cycles are influenced by the diurnal humidity fluctuations reported by Colin Pittendrigh, a founder of chronobiology, in his work on malaria-transmitting mosquitoes[19]. We also anticipate that our study in plants may lead to more in-depth investigation of humidity influences on mammalian physiology, as multiple studies have reported a correlation between humidity and disease susceptibility in humans[56–58].

## Methods

**Materials and growth conditions**. *LHYp:LUC* and *TOC1p:LUC* lines in WT and in *prr7-3 prr9-1* were generated by Salomé and McClung.[30] whereas the *CCA1p: LUC* line was described in Pruneda-Paz et al.[59]. The *cca1 lhy toc1* triple mutant was provided by Mizuno, T. and generated in Ito et al.[60] and the *35S:RRS1* transgenic line was provided by Jones, J. D. G. and generated in Sarris et al.[61]. Plants were grown in parallel in two identical Percival chambers (AR models) where light, temperature and humidity could be controlled independently. Light provided by Philips AltoII tubes was set at 100 μmol m$^{-2}$ s$^{-1}$ during the day or LL and 0 μmol m$^{-2}$ s$^{-1}$ at nights; temperature and air humidity were set according to experimental designs. Plants were grown in MetroMix 360 soil in 72-well flats (Hummert # 11–0700) and watered every 2–3 days to keep the soil volumetric water content (soil moisture) between 20% and 40% so that they did not experience drought stress. When applicable, plant genotypes were randomized to minimize position-specific effects and conditions in Percival chambers were switched between experiments (e.g., 50% constant RH versus 50%/90% RH oscillating humidity) to minimize chamber-specific effects. When three humidity conditions were needed, e.g., when comparing leaf-rosette biomass accumulation (Fig. 1a, b), plants were exposed to oscillating humidity by manually moving them between the 50% RH chamber and the 90% RH chamber.

**Luciferase imaging chamber and assay conditions**. To simultaneously image luciferase reporter activities in two sets of soil-grown mature plants (up to 192) under constant and oscillating humidity, respectively, while controlling light and temperature, we custom-built an imaging chamber with two compartments. Humidity was fed to the chamber by ultrasonic humidifiers (one Holmes humidifier for each compartment) and humidity in the room was kept at 50% RH using a dehumidifier (General Electric). Light was provided by red and blue light-emitting diode lights in the ratio of 2 to 1, respectively, and the fluence rate set at 100 μmol m$^{-2}$ s$^{-1}$. Temperature fluctuations in the chamber were kept within 1 °C by providing a constant flow of air at 22 °C into the chamber with and without the mist. Three-week-old transgenic plants carrying luciferase reporters were sprayed with 2.5 mM luciferin (Gold Biotechnology) in 0.02% Triton X-100 (Sigma) at both 24 and 12 h before luciferase imaging. Plants were then placed into the custom-built imaging chamber under specified light and humidity conditions and images were obtained using a charge-coupled device camera (PIXIS 2048) at 20 min exposure time every 2 h. Throughout the luciferase imaging, temperature, soil moisture, air humidity and light intensity were recorded every minute by Vernier probes STS-BTA, SMS-BTA, RH-BTA and LS-BTA, respectively. For the humidity entrainment experiments involving luciferase imaging, 50% RH was used as the free-running condition because it was technically more plausible to maintain constant 50% RH than 70% RH in the imaging chamber. Leaf temperature was measured in the Percival chamber using a FLIR A320sc IR imaging camera.

**Periodicity analysis and synchronization index**. To monitor clock gene expression, 3-week-old plants carrying luciferase reporters were sprayed with 2.5 mM luciferin 24 and 12 h before being moved into the imaging chamber. In the imaging chamber, plants were kept under their growth conditions for at least 1 day to stabilize gene expression before imaging at 2-h intervals. Gene expression was determined by bioluminescence emission of the area around the shoot apical meristem. For each plant, luciferase data were normalized to its mean expression value or its peak expression before humidity treatment and periodicity analysis was performed using the Biodare's FFT-NLLS and LS tools[62,63] with an expected period

between 18 and 36 h. Rhythms with relative amplitude error ≤ 0.7 were considered rhythmic and theirs calculated amplitudes and circadian phases by averaged peaks relative to time zero were reported. Supplementary Table 2 contains periodicity analysis using the Biodare's LS tool and the synchronization index, which was calculated using the formula:

$$\text{Sync.} = \frac{1}{N} * \left| \sum_{k}^{N} (\cos \theta k + i \sin \theta k) \right|$$

with θk the calculated circadian phase and N the number of oscillators.

**RNA extraction and quantitative PCR**. Plant RNA extraction was performed according to protocols provided by the manufacturers of the TRIZOL reagent (Ambion) and the Direct-zol (Zymo). cDNA construction was performed using superscript III reverse transcriptase enzyme (Invitrogen); and quantitative PCR (SYBR Green PCR kit, Roche) was carried out according to the manufacturers' protocols. Total bacterial RNA was extracted using the RNeasy kit (Qiagen) and the Bacteria Protect Reagent (Qiagen) following the manufacturer's instructions. Briefly, bacteria were re-suspended, washed twice in cold phosphate-buffered saline containing the Bacteria Protect Reagent and mechanically disrupted by 10 min of sonication (Bioraptor300). RNA was purified from the extract using the RNeasy kit and DNA was removed by treating total RNA with two rounds of the Ambion DNase I enzyme. cDNA was synthesized using random hexamers (Roche) and superscript III reverse transcriptase enzyme (Invitrogen) according to the enzyme protocol. Primer pairs (Supplementary Table 3) were selected to amplify regions of approximately 120 bp. Plant housekeeping gene *UBIQUITIN 5* (*UBQ5*) and bacterial genes *gyrA* and *gap1* were used as internal controls. When indicated, experimental replicates were combined using a linear mixed effect model (lme4 package in CRAN) before significant differences were determined by Student's *t*-test or analysis of variance (ANOVA). Results are reported as mean ± s.d. of three independent experiments.

**Fitness-related trait measurements**. Plants derived from a single parent were grown in soil under the indicated conditions and randomized in 72-well flats with respect to their genotypes to minimize position-specific effects. Conditions in Percival chambers were switched between experiments to minimize chamber-specific effects. Leaf rosettes from three plants were harvested together 3 weeks after germination. Fresh weight was measured immediately after harvest and dry weight was measured after drying at 65 °C for 7 days. Flowering time was reported as days to bolt (1 cm inflorescence). For seed mass and seed count measurements, plants were watered until leaf senescence was visible to at least 50% of leaves and then left to dry. Seeds from at least 12 plants were pooled and stored in a paper bag for at least 2 weeks before seed mass was measured using an analytical balance. Per experiment and condition, three samples of dry seed pools were weighted and seeds in each pool were counted to determine the average seed mass. Experimental replicates of seed mass and seed count were combined using a linear mixed effect model (lme4 package in CRAN) before significant differences were determined by Student's *t*-test or ANOVA. Results are reported as mean ± s.d. of three independent experiments.

**Pathogen infection**. Whole leaves of 3-week-old soil-grown plants were infiltrated with *Pst* (OD$_{600nm}$ = 0.001), *Pst AvrRpt2* (OD$_{600nm}$ = 0.001) or *Pst hrpL*(−) (OD$_{600nm}$ = 0.01) in 10 mM MgSO$_4$ and tissue collected 3 days after inoculation. Colony-forming units were counted according to Durrant et al.[64].

**Ion leakage measurement**. Ion leakage was determined by measuring electrolytes leaked from leaves inoculated with *Pst AvrRpt2* (OD$_{600nm}$ = 0.001) or *Pf AvrRps4* (OD$_{600nm}$ = 0.02). Ion leakage measurements were performed at 36 h post inoculation when ETI-associated PCD started to occur. Similar to the method described in Woo et al.[65], two leaves per plant were immersed in 5 mL of 400 mM mannitol with gentle shaking for 2 h, after which the sample conductivity was measured using Orion Star series meter (Thermo scientific) and normalized to total conductivity determined after boiling samples for 10 min. Six samples per growth condition were reported collectively as mean ± s.d. The experiment was repeated three times with similar results.

**Statistical analysis**. Statistical analyses were performed using GraphPad Prism 6 or the R programming language (R version 3.3.1). All centre values shown in figures are means of biological or experimental replicates. When mixed linear model was used, statistical analysis was performed on the mean estimates and their standard deviations.

## Data availability

The data that support the findings of this study are available in Supplementary Data 1 and in the Biodare repository.

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

## Acknowledgements

We acknowledge Dr. W. Wang for proposing the hypothesis that humidity is a plant zeitgeber. We thank Dr. X-D. Fu and Dr. H. Li for their help in surveying the influence of humidity on all clock genes; Dr. M. Dickey and Dr. J. Genzer for lending us the FLIR A325sc camera used to monitor leaf temperature; J. Motley, S. Zebell, Dr. P. Zwack and Dr. R. Zavaliev for commenting on this manuscript. We also thank the Millar lab for hosting data related to this manuscript in the Biodare repository. This work was supported by grants from the National Institutes of Health (NIH) (1R01-GM099839-01, 2R01-GM069594-09 and 5R35-GM118036) and by the Howard Hughes Medical Institute and the Gordon and Betty Moore Foundation (through grant GBMF3032) to X.D., a Defense Advanced Research Projects Agency (DARPA) Biochronicity Grant (DARPA-BAA-11-66), NIH Director's New Innovator Award (DP2 OD008654-01) and Burroughs Wellcome Fund CASI Award (BWF 1005769.01) to N.E.B.

## Author contributions

M.M. and X.D. designed the experiments. M.M., L.L., S.K, J.M. and E.M.M. performed the experiments and M.M. performed statistical analyses. N.E.B. and X.D. supervised the project. M.M., S.K., N.E.B. and X.D. wrote the paper with input from all co-authors.

## Additional information

**Competing interests:** The authors declare no competing interests.

