## [Peer Review File · Nature Communications]

Reviewers' comments:

Reviewer #1 (Remarks to the Author):

This paper addresses an intriguing concept of humidity as a zeitgeber and is indeed interesting. The authors propose that the natural oscillation of humidity serves as a zeitgeber in plants and simulate the humidity oscillations measured in Harvard Forest (geographical data missing) during spring and autumn equinoxes in their experiments. They demonstrate that plants accumulate more biomass when humidity oscillates than when it is constant. They also show that plants with compromised clock function show no difference in biomass accumulation in constant or oscillating humidity, even in the presence of light:dark cycles. They then go on to carry out experiments to investigate whether humidity can act as zeitgeber and its effect on aspects of clock function. Overall this is an interesting paper with novel findings that is likely to be influential in the field of chronobiology in general.

The following points need to be addressed:

1. Abstract line 33-35 "Indeed, in the presence of both zeitgebers as in nature, light sets the phase of the circadian clock whereas humidity reinforces the clock to further increase plant fitness." There are more than two zeitgebers in nature – should be re-written. It is not clear what is meant by "reinforces the clock".
2. Abstract lines 35-36 "Plant immunity is also enhanced at night when humidity naturally rises, likely to counter the increased pathogen virulence." It needs to be made clear that ETI was enhanced at night, and not overall immunity. Work by Bhardwaj et al (2011) and Zhang et al. (2013) show that plant immunity (PTI) to *Pseudomonas syringae* is actually enhanced in the morning, and diminished at night. This line should be rewritten to make it clear which pathogen is being referred to as this has implications for the defence response and role of humidity, and clarifying these two points.
3. P5 line 105 – the authors state that the plants were grown in the purpose built imaging chamber where humidity was controlled, and then state that "After 3 weeks, plants were moved to the imaging chamber". Please clarify.
4. It is not clear which reporter lines were used in these experiments. The authors need to clarify exactly which lines were used and cite the correct paper(s). The paper Salomé, P. A., and McClung, C. R. (2005). PRR7 and PRR9 are partially redundant genes essential for the temperature responsiveness of the Arabidopsis circadian clock. *Plant Cell* 17, 791–803 describes the production of CCA1p:luc, LHYp:luc and TOC1p:luc lines and shows their expression profiles. The paper referred to: Kim, W.Y., Salomé, P.A., Fujiwara, S., Somers, D.E. & McClung, C.R. Characterization of pseudo-response regulators in plants. *Methods Enzymol* 471, 357-78 (2010), describes the work in Salomé and McClung (2005) and again shows their profiles. The paper by Pruneda-Paz et al. (2009) that is also referred to describes the production of a different CCA1p:luc line.
5. The LHYp:luc and TOC1p:luc lines created by Salomé and McClung (2005) have an approximate 10-fold lower level of Luc expression than the CCA1p:luc line (visible in both Salomé and McClung (2005) and Kim et al., (2010). This detection difference may have contributed to the authors noting some effects e.g. on phase shifting in the CCA1p:luc line and not the other two, and attributing the apparent differences to signalling differences.
6. The rhythms in Fig 2 and supp Fig 3 are intriguing. The main text (p5 lines 101-103) states that "From seed germination, we grew two populations of plants in LL at 22°C with one population under constant 50% RH whereas the other population under 12 h 50% | 12 h 90% oscillating RH". Although

the oscillations in constant humidity look almost arrhythmic or not very robust (Fig 2 and supp Fig 3) with no clear phase, there is evidence of circadian rhythmicity in some individual lines (and periods are extracted). What is responsible for the rhythms in non-humidity-entrained plants?

7. "Gene expression was normalized to the peak expression between time -26 h and -4 h." This is an odd practice – please explain why this was done.

8. How do the authors explain the phase difference in LHY and CCA1 expression observed in plants entrained to different humidity periods (Fig 3) and the absence of response of LHY to a humidity pulse (Fig 4)?

9. The experiments in Fig 4 g and h would have been better to show TOC1 or CCR2 expression rather than CCA1. It does look like LHY expression is rhythmic in 50% humidity in 4h. The reviewer suggests that the experiment shown in figure 1 should be repeated with single mutants of *cca1*, *lhy*, and *toc1* to demonstrate which component is key for humidity's zeitgeber effects.

10. P9 lines 183-184 – "...*Pseudomonas syringae* against which the circadian regulation of plant immunity has been most studied 33-35" authors should include reference 36 (Zhang et al, (2013) and the study by Bhardwaj et al. (2011) which both studied *Pseudomonas syringae*.

11. While pressure inoculation bypasses stomatal regulation (Zhang et al, (2013) and Bhardwaj et al. (2011)), it would be interesting to understand the effect of the oscillation of humidity on the stomatal entry and PTI by these bacterial pathogens, and thus include the spray inoculation as used in Zhang et al ref 36.

12. P9 lines 192-195 "However, it significantly enhanced ETI elicited by Pst carrying the AvrRpt2 effector (Pst AvrRpt2) in the evening-inoculated leaves (Fig. 6d), but not in morning-inoculated leaves (Supplementary Fig. 6c), suggesting a time-of-the-day sensitivity to ETI induction." Would be better to have these data in one figure, please. Similarly for Fig. 6e and Supplementary Fig 6d.

13. P2 line 42 of supplementary material "This experiment lasted for 5 h and measurement were recorded at every 30 min" referring to leaf temperature measurement only?

14. Supp fig 7 : axis label states 95% RH and main text states 95% RH, but legend refers to 90% RH – please clarify

Reviewer #2 (Remarks to the Author):

Mwimba et al investigate the influence of humidity on circadian clock, and posit that humidity is another factor that entrains the clock. This is a very interesting hypothesis, and the authors have made substantial efforts to address this question. They tested for the effects of oscillating humidity versus consistent humidity on parameters related to plant physiology and plant-microbe interactions. In order to independently control humidity, temperature and light, the authors had to design a custom growth chamber, so these were not trivial experiments to conduct. We applaud the authors' efforts to dissect the contribution of humidity to plant physiology and plant-microbe interactions, however we did not feel that the data supported the conclusions.

Major comments:

The authors' primary conclusion is that humidity serves as a zeitgeber (i.e., resetting cue) for the circadian clock. This would be an exciting discovery. The authors present several experiments in

Figures 2-4 to support this idea. However, these do not directly address this question. Figure 3 shows that long days (produced by 16 h 50% humidity followed by 8h 90% humidity) cause a delay in the phase of CCA1p:LUC and LHYp:LUC relative to day neutral conditions. This change in phase does not appear for TOC1p:LUC expression. Therefore, it appears that whatever the effect of humidity on CCA1 and LHY expression, this does not influence the phase of the clock itself. Instead, the effects are limited to CCA1 and LHY. The humidity pulse experiment in Figure 4 is closest to a direct test of humidity as a zeitgeber, but the outcome is not consistent with humidity globally resetting the clock. Brief application of an authentic zeitgeber will change the phase of the entire clock system, both core clock components and outputs. While Figure 4a shows a humidity pulse greatly elevates and delays CCA1p:LUC expression (Figure 4a, b), this treatment has no apparent effect on the phase of LHYp:LUC and TOC1p:LUC expression (Figure 4c, e). A similar experiment with known zeitgebers, such light or temperature, would produce a long-term change in clock phase that would be evident for all clock-regulated genes. The direction and magnitude of the phase change caused by zeitgeber application would depend on the time of day the pulse is administered. This resetting behavior is evident in a phase response curve (examples in Covington et al. 2001 Cell, Salome and McClung 2005 The Plant Cell, and Thines et al. 2010 PNAS) or phase transition curve (example in Locke et al. 2005 EMBO). The authors must do one of these tests to conclusively show that humidity is a zeitgeber for the clock. Given the antiphase nature of expression driven by humidity cycles, it will be interesting to see how the shape of the PRC/PTC compares to those established for light and temperature.

The authors conclude that cycles in humidity conditions drive rhythmic expression of circadian clock genes, CCA1 expression rises acutely with a shift to high humidity conditions, and humidity cycles mimicking natural conditions elevate the amplitude of diurnal rhythms for CCA1, LHY, and TOC1 expression. All this work was done with luciferase reporter constructs. The authors have not ruled out the possibility that luciferase reporters are influenced by humidity differently from actual clock gene expression. It is important for the authors to use a second method, such as qPCR, to confirm that humidity is actually changing gene expression as shown by the reporters. They need not repeat all the time courses; instead, they should confirm that reported changes in amplitude are evident with another method.

Line 106: The authors have yet to show that "humidity oscillation could entrain the circadian clock and set the phases of..." This is an appropriate statement only after the authors have presented additional evidence to support this conclusion.

Line 152 and Figure 4g, h: How do the authors rationalize the observation that the *cca1* mutant shows correct phasing of LHY expression in response to humidity? These data indicate that the humidity effects on LHY expression do not at all depend on CCA1.

Line 177: Another interpretation of the observations in Figure 6 is that arrhythmic clock mutants will always be compromised in these physiology tests, no matter the conditions.

The authors use the *prp7 prp9* mutant (Sup Fig 4) to conclude that humidity is important, not temperature. The period and phase values in panel c are comparable between 50% humidity and humidity cycles; the rhythms are just less organized with 50% humidity. It is possible that cryptic temperature cycles contribute to the rhythms in the *prp7prp9* background. Salome and McClung (Plant Cell 2005) demonstrate that the double mutant has driven rhythms in temperature cycles, but not in constant conditions (Figure 6 in that paper).

The major finding of Figure 6 (and Supplementary Figure 6) is that AvrRpt2-induced ETI is enhanced in plants grown under natural humidity oscillations, but only when plants are inoculated in the evening (ZT13), not in the morning (ZT1). Given that ZT13 precedes a period of 90% humidity, this

observation is unexpected in light of the wealth of data indicating that high humidity suppresses programmed cell death (PCD) responses arising from either ETI or autoimmunity. In addition, Xin et al. (ref 43) noted that at high humidity, AvrRpt2-associated ETI blocked the water-soaking symptoms caused by PstDC3000, suggesting that ETI would not be enhanced by increased local water availability. With regards to timing, the authors' observations also contradict previously documented circadian rhythms, such as the finding that genes associated with RPP4-mediated PCD exhibit peak expression at dawn in a CCA1-regulated manner, and that plants are more susceptible to Hpa infection when inoculated at dusk versus dawn (Wang et al., 2011; Nature 470:110-115). A reduced capacity to respond to *P. syringae* in the evening versus the morning has also been suggested (Bhardwaj et al., 2011 [PLoS One 6:e26968]; Zhang et al., 2013 [PLoS Pathogens 9:e1003370]). Greater clarity on the humidity/circadian relationship would be provided by bacterial growth assays that include plants maintained at a constant 90% RH. This would allow a comparison of the relative influence of high humidity versus the timing of inoculation on ETI.

The choice of AvrRpt2 also may have influenced the results obtained from the ETI experiments. AvrRpt2 is known to modulate auxin signaling, and auxin responsiveness appears to be gated by the circadian clock (Covington and Harmer, 2007; PLoS Biol 5:e222). There is some suggestion that auxin can affect ETI (discussed in Naseem et al., 2015; J Exp Bot 66:4885:4896), although this remains to be conclusively demonstrated. Nonetheless, it would be worth repeating the growth assay and ion leakage experiments with a different ETI-eliciting effector to rule out any potentially confounding effects of other circadian-regulated components.

Supplementary Figure 7: while the observation of type III induction by high humidity is interesting, it is not clear how this relates to the results shown in figure 6, in which all bacteria are infiltrated into the leaf apoplast as liquid suspensions (i.e. maximum humidity). Furthermore, if increased effector expression accounted for the differences in growth, then increased growth of Pst in Col-0 and Pst+AvrRpt2 in *rps2* mutants might be expected under natural humidity oscillations. This may be masked by the high levels of bacterial growth observed by day 3, in which case an earlier time point should be analyzed. At this point, the connection between humidity, effector expression levels, and ETI in the current manuscript appears to be overly simplistic, especially given the literature referenced above as well as the documented circadian fluctuations of other molecules such as salicylic acid and indicators of cellular redox state.

Materials and Methods Line 53: Figures 2-4, estimated period is calculated with FFT-NLLS. The standard approach with this tool is to apply a cutoff for the relative amplitude value produced by this analysis to eliminate arrhythmic luciferase traces. The authors need to indicate what they did for their analysis.

It is also not clear for each set of experiments what conditions the plants experienced prior to the time window shown in the figure. This seemed to change for each Figure, but was difficult to determine. The nature of the prior treatment can have a significant effect on the rhythms and clock resetting behavior.

Minor comments:

Line 55: change to "...circadian clocks alluded to effects of light, temperature, and air humidity on circadian clock regulation."

Figure 1: Based on the figure legend, the y-axis of figure 1b should read: "Rosette weight (mg)" and figure 1c should read: "Rosette dry weight (mg)".

Depending on the figure, error bars are standard deviation at times or standard error of the mean. A consistent method to represent uncertainty should be used throughout, preferably the more informative standard deviation.

line 416: "black" should read: "blue"

line 452: the legend mentions a black arrow, but no arrows appear in figure 5. Please also describe the growth conditions prior to the application of the natural humidity oscillation.

Line 461 and 480: abbreviation for standard error of the mean is missing the "m".

Figure 6: Based on the figure legend, the y-axis of figure 6a should read: "Rosette dry weight (mg)".

Supplementary Material, line 34: "CDD" should read: "CCD"

Supplementary Figure 7: Please indicate the genes whose fold change is statistically significant from one-fold. The error bars for hopD1 and HopN1 in HMM appear to overlap with the one-fold cutoff. Also, it would be helpful to know the fold-change of each gene for HMM versus KB media (at 50% RH). This would reveal the relative contribution of signals from the media versus humidity cues with regards to gene induction.

Reviewer #3 (Remarks to the Author):

The paper presents a very interesting and worthwhile investigation of a potentially new input to the clock system. RH is also tested for its broader effects on Arabidopsis physiology and pathogen response, with promising results. The context from Harvard Forest met data is welcome. The studies are well-conceived, novel and original but raise significant technical issues that are not fully addressed. The present set of results are therefore more strongly interpreted in the text conclusions than seems warranted, and either the text or the results presented should be updated.

1. The mechanism and physiological importance of RH.

a. Leaf temperature clearly was warmer at 90% than 50% RH by about 1C (Supple Fig 2), as expected from greater transpirational cooling of the well-watered plants into the drier air at 50%. Stomatal dynamics in particular conditions will obviously affect this. If higher RH is associated with higher leaf temperature, this is sufficient to explain the sequence of gene expression peaks before/during/at the end of the high RH intervals in Figures 2 and 3. This would not be 'antiphase entrainment' as suggested in lines 124&ff, it is very similar to previous experimental results in temperature cycles (many references cited). It does result in clock entrainment in antiphase to the natural RH cycle, but the study was not conducted under natural conditions. This might also explain why Fig 4g-h show that RH entrainment continues in *cca1* mutants: temperature is known to affect many clock components, even if RH principally affected CCA1 expression.

b. The measured 1C change is small, but this measurement was in a separate study in a long interval at 50% or 90% RH. The temperature cycle during an RH cycle might be larger, at least transiently. Repeated, transient signals are clearly sufficient for entrainment in plants. Low amplitude temperature is also sufficient, as another example, Fujiwara, S. et al. *Plant Cell* 20, 2960-2971, doi:10.1105/tpc.108.061531 (2008) show that a 4C temperature cycle entrains sufficiently for well-coordinated rhythms even at the level of pooled RNA samples, a much less sensitive assay than the single-apex LUC measurements here or the seedlings tested in ref 27 Michael 2002. Smaller-amplitude temperature cycles are not known to be ineffective for entrainment in plants, simply no data are

available: the authors could test this, see point 3a. A repeated 1C temperature cycle is sufficient to entrain mammalian cells (Saini et al. Genes Dev 2012).

c. The *prp7 prp9* control is worthwhile, as indeed it might have failed to entrain to RH. However, this mutant was reported to alter entrainment to 22C/12C cycles but not to 22C/28C cycles (refs cited). It is therefore unclear whether it is 'temperature-irresponsive at 21-22°C' as claimed. There is no clear interpretation of its continued entrainment to RH, though the data are clear.

d. The response of *CCA1:LUC* in Fig 4 is far stronger than one would expect of a 1C temperature change, suggesting a specific RH input, but note the technical concern on the other genes below. Also, there is no temperature record for any of the plants in these studies.

Conclusion. Points a and b above raise the reasonable possibility, or one could argue most parsimonious, testable hypothesis, that the mechanism for applied RH changes to affect the clock is via a small change in leaf temperature. That does not mean that the RH entrainment is not important physiologically. The authors might discuss which environmental conditions would give RH a significant entraining role alongside light and ambient temperature. Point d in contrast argues for a distinct RH effect, albeit with technical caveats.

2. Physiological relevance

a. When RH is combined with LD cycles, the clock phase clearly follows LD alone, with no discernible RH effect on clock reporter phase in Fig. 5. The result would be the same for a very small temperature cycle.

b. Concluding that the RH cycle reinforces clock amplitude based on LUC alone is unsafe. The amplitude of LUC reporter rhythms is notoriously hard to interpret, as it depends on LUC-dependent factors (translation rate, and substrates O₂ and ATP) as well as the clock transcriptional regulators. The authors should note this possibility, as it will be harder to support their conclusion more directly. RNA assays in the simulated cycles might struggle to detect this change in amplitude. A transfer into constant conditions with some test of clock 'quality' (eg synchrony among LUC rhythms in individual plants) could be an alternative.

c. There is no control to test whether this effect is specific to the 'natural' phase relationship of LD and RH.

3. Technical concerns.

The discrepancies between clock markers in Figures 2 and 4 are hard to interpret or suggest variability somewhere in the assays.

a. The LL assay conditions are clearly effective, allowing for wide desynchronisation among individuals that persists after transfer to the imaging apparatus (as Fig 2 controls show). This gives the authors the opportunity to test whether applying a 1C (on average over 12h) cycle can entrain the clock.

b. In tests of novel entraining stimuli, a common way to eliminate possible confounding effects is by demonstrating antiphase entrainment, in response to antiphase input cycles, and stable, opposite phases in subsequent constant conditions. Fig 3 goes in this direction, with the proviso that all genes would be expected to behave similarly in LL.

c. If a changed RH cycle (Fig 3) or RH pulse (Fig 4) causes a phase shift in the clock that persists under constant conditions (as claimed in both figures), it is expected to alter all clock-controlled genes. Fig 3 and Fig 4 show a phase shift in only two (Fig 3) or one (Fig 4) clock gene, or more

specifically, in the transgenic lines carrying these reporters. This implies that the RH changes had a long-lasting effect on the coupling between the clock genes, such that some genes can change phase without affecting the others. During ongoing entrainment, such a change in waveform is well known. Under constant conditions, however, it is surprising and defies a simple biological explanation. Technical variability would be one possibility. It has been argued that such different behaviour among other reporters reflected control by distinct (often, anatomically separate) clocks. That seems unlikely for widely-expressed, core clock components. Lines 121-122 do not address this concern.

d. Line 229. I doubt the SAM alone was imaged in a field of view comprising hundreds of plants. The apical region typically has high luminescence signal from the young emerging leaves surrounding the SAM, and these have stomata.

4. Interpretation of c-l-t- clock mutants, Fig. 2 and 6. E.g. Line 87 "these effects were clock-dependent". These figures have valuable data but the specific phrase above is an overinterpretation, rather the effects depend on clock genes. Studies of a clock-less mutant (such as *cca1 lhy toc1*) cannot link any of the mutant's phenotypes uniquely to a defect in circadian timing. Deficiency in these three TFs affects multiple, non-rhythmic genes, which could result in the loss of biomass/responsiveness to changing humidity. Assays such as a T-cycle study of period mutants are required to show that the phenotype is specifically clock-related (rather than clock-gene-related), because a deficiency of timing can then be rescued predictably by a corresponding alteration in the environment. A larger panel of clock mutants under a single condition would be an intermediate step.

5. Minor points.

a. The 'natural humidity oscillation' should probably be rephrased, for example, to 'simulated natural conditions'.

b. The legend text and elsewhere should reference the FFT-NLLS method [Plautz et al. 1996], as this is materially different from what is described as a Fourier transform. The BioDare resource offers several phase-measurement options with FFT-NLLS, and the authors should specify which was used here.

c. I strongly support using the data repository to share data but users will retrieve a wide range of irrelevant, public data by logging in and performing the recommended search with "*". It would be more direct if the authors listed the particular experiments that correspond to their figures, in the supplement. Those records can then be made publicly accessible, using the "Open Access publish" function, so no login is required.

This review was provided by Andrew Millar, University of Edinburgh.

Reviewer #1

This paper addresses an intriguing concept of humidity as a zeitgeber and is indeed interesting. The authors propose that the natural oscillation of humidity serves as a zeitgeber in plants and simulate the humidity oscillations measured in Harvard Forest (geographical data missing) during spring and autumn equinoxes in their experiments. They demonstrate that plants accumulate more biomass when humidity oscillates than when it is constant. They also show that plants with compromised clock function show no difference in biomass accumulation in constant or oscillating humidity, even in the presence of light:dark cycles. They then go on to carry out experiments to investigate whether humidity can act as zeitgeber and its effect on aspects of clock function. Overall this is an interesting paper with novel findings that is likely to be influential in the field of chronobiology in general.

We thank Reviewer #1 for finding the manuscript “an interesting paper with novel findings that is likely to be influential in the field of chronobiology in general”.

1. Abstract line 33-35 “Indeed, in the presence of both zeitgebers as in nature, light sets the phase of the circadian clock whereas humidity reinforces the clock to further increase plant fitness.” There are more than two zeitgebers in nature – should be re-written. It is not clear what is meant by “reinforces the clock”.

The reviewer’s concern has been addressed by revising the text to: “In the presence of light and humidity cycles, however, light sets the phase of the circadian clock whereas humidity reinforces the clock rhythm to further increase plant fitness.” (Lines 31-33)

2. Abstract lines 35-36 “Plant immunity is also enhanced at night when humidity naturally rises, likely to counter the increased pathogen virulence.” It needs to be made clear that ETI was enhanced at night, and not overall immunity. Work by Bhardwaj et al (2011) and Zhang et al. (2013) show that plant immunity (PTI) to *Pseudomonas syringae* is actually enhanced in the morning, and diminished at night. This line should be rewritten to make it clear which pathogen is being referred to as this has implications for the defence response and role of humidity, and clarifying these two points.

To address the reviewer’s concern, we have changed the text to: “Moreover, effector-triggered immunity is enhanced at night with the rise in humidity ...” (Lines 33-34). In contrast to Bhardwaj et al (2011) and Zhang et al. (2013), this was not an experiment to compare ETI in morning- versus evening-inoculated plants. Instead, we looked at the influence of oscillating humidity on ETI in either morning- or evening-inoculated plants. We have modified the text to make this clearer (lines 207-217).

3. P5 line 105 – the authors state that the plants were grown in the purpose built imaging chamber where humidity was controlled, and then state that “After 3 weeks, plants were moved to the imaging chamber”. Please clarify.

Plants were first grown in Percival chambers under controlled conditions then transferred to the imaging chamber. We have rewritten the text to: “From seed germination, we grew plants in two Percival growth chambers set to LL 22°C conditions with humidity kept under constant 50% RH in one chamber whereas humidity was set to oscillate 12 h 50%| 12 h 90% RH in the

other chamber. After 3 weeks, plants were moved to the imaging chamber and clock gene expression was measured by luciferase reporter activities in the shoot apical meristem (SAM), which has been shown to be the central timekeeping organ for plants” (Lines 106-108). Also, we have included humidity and temperature measurements of the Percival chambers in the new Supplementary Fig. 2a,b to show that the environment is effectively controlled.

4. It is not clear which reporter lines were used in these experiments. The authors need to clarify exactly which lines were used and cite the correct paper(s). The paper Salomé, P. A., and McClung, C. R. (2005). PRR7 and PRR9 are partially redundant genes essential for the temperature responsiveness of the Arabidopsis circadian clock. *Plant Cell* 17, 791–803 describes the production of CCA1p:luc, LHYp:luc and TOC1p:luc lines and shows their expression profiles. The paper referred to: Kim, W.Y., Salome, P.A., Fujiwara, S., Somers, D.E. & McClung, C.R. Characterization of pseudo-response regulators in plants. *Methods Enzymol* 471, 357-78 (2010), describes the work in Salomé and McClung (2005) and again shows their profiles. The paper by Pruneda-Paz et al. (2009) that is also referred to describes the production of a different CCA1p:luc line.

We have addressed the reviewer’s concern by modifying the Materials section as follows: “LHYp:LUC and TOC1p:LUC lines in WT and in *prr7-3 prr9-1* were generated by Salomé, P. A., and McClung, C. R. whereas the CCA1p:LUC line was described in Pruneda-Paz et al. The *cca1 lhy toc1* triple mutant was provided by Dr. Mizuno and generated in Ito et al...” (Lines 289-292)

5. The LHYp:luc and TOC1p:luc lines created by Salomé and McClung (2005) have an approximate 10-fold lower level of Luc expression than the CCA1p:luc line (visible in both Salomé and McClung (2005) and Kim et al., (2010). This detection difference may have contributed to the authors noting some effects e.g. on phase shifting in the CCA1p:luc line and not the other two, and attributing the apparent differences to signalling differences.

The reviewer was right that the non-responsiveness of LHYp:LUC and TOC1p:LUC to the 6h 90% RH pulse of humidity might be due to their lower levels of expression. When we performed the 6 h 90% RH treatment using qRT-PCR, the expression of LHY mRNA was responsive to humidity as shown in the new Supplementary Fig. 7c.

6. The rhythms in Fig 2 and supp Fig 3 are intriguing. The main text (p5 lines 101-103) states that “From seed germination, we grew two populations of plants in LL at 22°C with one population under constant 50% RH whereas the other population under 12 h 50%| 12 h 90% oscillating RH”. Although the oscillations in constant humidity look almost arrhythmic or not very robust (Fig 2 and supp Fig 3) with no clear phase, there is evidence of circadian rhythmicity in some individual lines (and periods are extracted). What is responsible for the rhythms in non-humidity-entrained plants?

As reported by Salome, P.A., Xie, Q and McClung, C.R (2008), individual plants grown in the absence of entraining signals have intrinsic rhythmic expression of circadian clock genes (Figure 1D-F and Table 1 in that paper). However, in the absence of entraining signals, these oscillations are not synchronized between different plants.

7. “Gene expression was normalized to the peak expression between time -26 h and -4 h.” This is an odd practice – please explain why this was done.

We are interested in the humidity effects on amplitude and not just phase and period. Thus, we needed to set the baseline for the before-and-after humidity treatment comparisons using the normalized peak expression.

8. How do the authors explain the phase difference in LHY and CCA1 expression observed in plants entrained to different humidity periods (Fig 3) and the absence of response of LHY to a humidity pulse (Fig 4)?

In response to the reviewer’s concern #5, we have updated the text and stated that *LHY* expression can be responsive to humidity pulses (lines 155-156; new Supplementary Fig. 7). The lack of responsiveness of *LHYp:LUC* in the old Fig. 4 might be due to the low expression level of the reporter as suggested by the reviewer. It is also possible that there might be a real difference in sensitivity to humidity between CCA1 and LHY which might explain the phase differences in *LHYp:LUC* and *CCA1p:LUC* in Figure 3.

9. The experiments in Fig 4 g and h would have been better to show TOC1 or CCR2 expression rather than CCA1. It does look like LHY expression is rhythmic in 50% humidity in 4h. The reviewer suggests that the experiment shown in figure 1 should be repeated with single mutants of *cca1*, *lhy*, and *toc1* to demonstrate which component is key for humidity’s zeitgeber effects.

The reviewer was likely misled by the title of the old Fig. 4 h, which showed the synchronization of *LHY* mRNA expression in the *cca1* mutant. Successful synchronization of *LHY* expression in WT (old Fig. 4g) and in the *cca1* mutant (old Fig. 4h) was used to show that besides CCA1, humidity targets other clock genes to synchronize the clock. Panels of old Fig.4 are now in Fig.3 and Supplementary Figs. 4 and 7.

10. P9 lines 183-184 – “...*Pseudomonas syringae* against which the circadian regulation of plant immunity has been most studied 33-35” authors should include reference 36 (Zhang et al, (2013) and the study by Bhardwaj et al. (2011) which both studied *Pseudomonas syringae*.

These citations have been added (Line 199).

11. While pressure inoculation bypasses stomatal regulation (Zhang et al, (2013) and Bhardwaj et al. (2011)), it would be interesting to understand the effect of the oscillation of humidity on the stomatal entry and PTI by these bacterial pathogens, and thus include the spray inoculation as used in Zhang et al ref 36.

It is known that stomatal opening is regulated by the circadian clock and by humidity directly, which could confound our results. Bypassing the stomatal control using infiltration allows us to have a more straightforward readout of the effect of the humidity-affected circadian clock on immunity. We have included this rationale in the main text (lines 199-203).

12. P9 lines 192-195 “However, it significantly enhanced ETI elicited by Pst carrying the AvrRpt2 effector (Pst AvrRpt2) in the evening-inoculated leaves (Fig. 6d), but not in morning-inoculated

leaves (Supplementary Fig. 6c), suggesting a time-of-the-day sensitivity to ETI induction.” Would be better to have these data in one figure, please. Similarly for Fig. 6e and Supplementary Fig 6d.

We deliberately kept the results of morning-infected experiment in the old Supplementary Fig. 6 to avoid the implication that pathogen growth in the evening-inoculated and the morning-inoculated leaves could be compared. These experiments were not carried simultaneously and therefore are not comparable. The main point of this experiment was to compare defense under constant humidity (70%) versus oscillating humidity (50%|90%).

13. P2 line 42 of supplementary material “This experiment lasted for 5 h and measurement were recorded at every 30 min” referring to leaf temperature measurement only?

Yes. The text has been edited to: “Leaf temperature experiment lasted 5 h...”

14. Supp fig 7 : axis label states 95% RH and main text states 95% RH, but legend refers to 90% RH – please clarify

The legend has been corrected to 95% RH. Thank you.

Reviewer #2

Mwimba et al investigate the influence of humidity on circadian clock, and posit that humidity is another factor that entrains the clock. This is a very interesting hypothesis, and the authors have made substantial efforts to address this question. They tested for the effects of oscillating humidity versus consistent humidity on parameters related to plant physiology and plant-microbe interactions. In order to independently control humidity, temperature and light, the authors had to design a custom growth chamber, so these were not trivial experiments to conduct. We applaud the authors’ efforts to dissect the contribution of humidity to plant physiology and plant-microbe interactions, however we did not feel that the data supported the conclusions.

We thank Reviewer #2 for finding our discovery exciting and applauding our “effort to dissect the contribution of humidity oscillation to plant physiology and plant-microbe interactions”.

The authors’ primary conclusion is that humidity serves as a zeitgeber (i.e., resetting cue) for the circadian clock. This would be an exciting discovery. The authors present several experiments in Figures 2-4 to support this idea. However, these do not directly address this question. Figure 3 shows that long days (produced by 16 h 50% humidity followed by 8h 90% humidity) cause a delay in the phase of CCA1p:LUC and LHYp:LUC relative to day neutral conditions. This change in phase does not appear for TOC1p:LUC expression. Therefore, it appears that whatever the effect of humidity on CCA1 and LHY expression, this does not influence the phase of the clock itself. Instead, the effects are limited to CCA1 and LHY.

Under constant light conditions, humidity entrainment of all three core clock genes and three clock output genes were demonstrated in Fig. 2 in which there was a 12-h phase shift for all these genes compared to what is observed in nature (i.e., CCA1 and LHY peaked at the

transition from low-to-high humidity and *TOC1* peaked at the transition from high-to-low humidity). And the rhythms persisted under free-running conditions. In Fig. 3, *TOC1p:LUC* was not shifted in the 16 h 50%|8 h 90% RH regimen because the transition from high-to-low humidity was kept at the same time as in the day-neutral regimen. It was expected that phases of *CCA1p:LUC* and *LHYp:LUC* were delayed with the transition from low-to-high humidity transition whereas the phase of *TOC1p:LUC* was unchanged. Similar results were observed when plants were entrained to light using different photoperiods. For example, in Green et al. (2002) *Plant Phys.*, morning and evening genes peaked at their respective transitions (D/L and L/D; Figs. 2 and 3).

The humidity pulse experiment in Figure 4 is closest to a direct test of humidity as a zeitgeber, but the outcome is not consistent with humidity globally resetting the clock. Brief application of an authentic zeitgeber will change the phase of the entire clock system, both core clock components and outputs. While Figure 4a shows a humidity pulse greatly elevates and delays *CCA1p:LUC* expression (Figure 4a, b), this treatment has no apparent effect on the phase of *LHYp:LUC* and *TOC1p:LUC* expression (Figure 4c, e). A similar experiment with known zeitgebers, such light or temperature, would produce a long-term change in clock phase that would be evident for all clock-regulated genes. The direction and magnitude of the phase change caused by zeitgeber application would depend on the time of day the pulse is administered. This resetting behavior is evident in a phase response curve (examples in Covington et al. 2001 *Cell*, Salome and McClung 2005 *The Plant Cell*, and Thines et al. 2010 *PNAS*) or phase transition curve (example in Locke et al. 2005 *EMBO*). The authors must do one of these tests to conclusively show that humidity is a zeitgeber for the clock. Given the antiphase nature of expression driven by humidity cycles, it will be interesting to see how the shape of the PRC/PTC compares to those established for light and temperature.

Jürgen Aschoff, one of the founders of chronobiology, described a zeitgeber as a cue that could synchronize and entrain the circadian clock (Aschoff, 1954). Light, temperature and feeding cycles all satisfy these two criteria (De Candolle, 1833; Richter, 1922; Shirley, 1928; Reid and Finger, 1955). Therefore, we believe that our entrainment experiments (Figs. 2 and 3), in which we showed that humidity could synchronize the clock and reset the phase, are direct demonstrations that humidity oscillation is a zeitgeber.

Even though PRC (suggested by the reviewer) can be a powerful tool to investigate properties of circadian rhythms, it is not a direct demonstration *per se* that a cue is a zeitgeber. To our knowledge, only Hayden et al. (2013) used PRC to establish that a cue (sugar) “functions in entrainment”. Hayden et al. likely used PRC, instead of direct entrainment, because of the difficulty in oscillating the sugar content in the plant growth media.

The authors conclude that cycles in humidity conditions drive rhythmic expression of circadian clock genes, *CCA1* expression rises acutely with a shift to high humidity conditions, and humidity cycles mimicking natural conditions elevate the amplitude of diurnal rhythms for *CCA1*, *LHY*, and *TOC1* expression. All this work was done with luciferase reporter constructs. The authors have not ruled out the possibility that luciferase reporters are influenced by humidity differently from actual clock gene expression. It is important for the authors to use a second method, such as qPCR, to confirm that humidity is actually changing gene expression as shown by the reporters. They need not repeat all the time courses; instead, they should confirm that reported changes in amplitude are evident with another method.

We agree with the reviewer that it is necessary to rule out a possible artifact of using luciferase reporters. This was the reason that we performed qRT-PCR in the old Fig. 4g (the new Supplementary Fig. 4), which validated the luciferase results in Fig. 2. Moreover, we performed qRT-PCR in the new Supplementary Fig. 7, which showed that *CCA1* and *LHY* are both sensitive to a single 6-h humidity treatment. Additionally, qRT-PCR in the new Supplementary Fig. 8 showed higher expression of *CCA1* and *LHY* at dawn in plant grown under simulated natural conditions (LD, 50%|90% RH) compared to LD 70% RH, which validated the luciferase results in the old Fig.5.

Line 106: The authors have yet to show that “humidity oscillation could entrain the circadian clock and set the phases of...” This is an appropriate statement only after the authors have presented additional evidence to support this conclusion.

As stated above, we have definitive results that support this conclusion. In Fig. 2, we showed the active entrainment of the circadian clock, as demonstrated by the synchronization of clock genes during humidity cycles as well as after plants were released to free-running conditions. In Fig. 3, we used two different regimens of humidity cycles to demonstrate that the phases of *CCA1* and *LHY* were actively set by the low-to-high humidity transition whereas the phase of *TOC1* was set by the high-to-low humidity transition.

Line 152 and Figure 4g, h: How do the authors rationalize the observation that the *cca1* mutant shows correct phasing of *LHY* expression in response to humidity? These data indicate that the humidity effects on *LHY* expression do not at all depend on *CCA1*.

Based on these results, we could, at most, conclude that other components of the clock besides *CCA1* must be targeted by humidity. We stated this in our original manuscript in Line 152-153. Based on the architecture of the clock (*LHY* is a transcriptional target of *CCA1*), the *LHY* expression is expected to be affected by *CCA1* and other humidity-responsive clock component(s).

Line 177: Another interpretation of the observations in Figure 6 is that arrhythmic clock mutants will always be compromised in these physiology tests, no matter the conditions.

It is incorrect to assume that the clock is arrhythmic in the *cca1 lhy toc1* mutant because our experiments in Figure 6 were performed under LD conditions instead of LL, where the mutant was shown to be arrhythmic in Niwa et al., (2007). Moreover, we are not directly comparing WT with *cca1 lhy toc1* as the reviewer implied here. We are comparing how they each respond to constant (70%) versus oscillating (50%|90%) humidity.

The authors use the *prr7 prr9* mutant (Sup Fig 4) to conclude that humidity is important, not temperature. The period and phase values in panel c are comparable between 50% humidity and humidity cycles; the rhythms are just less organized with 50% humidity. It is possible that cryptic temperature cycles contribute to the rhythms in the *prp7prp9* background. Salome and McClung (Plant Cell 2005) demonstrate that the double mutant has driven rhythms in temperature cycles, but not in constant conditions (Figure 6 in that paper).

The lack of organization (i.e., asynchrony) is what we expected in the absence of a zeitgeber at constant 50% humidity. The observed sustained synchrony after the plants were released from oscillating humidity to the free-running conditions (which was not observed in Salome et al., Fig. 5) suggests that cryptic temperature cycles are less likely the source of entrainment in humidity oscillation conditions.

To further address the reviewer's concern, we performed a temperature entrainment experiment using the 12h 21°C|12h 22.5°C cycle which contains the temperature changes recorded due to humidity variations shown in Supplementary Figure 2. We found that WT was weakly entrained by the temperature changes and the *prp7 prp9* mutant was not entrained at all (new Supplementary Fig. 6). This new experiment further ruled out the possibility that the humidity entrainment in *prp7 prp9* was due to cryptic temperature cycles.

The major finding of Figure 6 (and Supplementary Figure 6) is that AvrRpt2-induced ETI is enhanced in plants grown under natural humidity oscillations, but only when plants are inoculated in the evening (ZT13), not in the morning (ZT1). Given that ZT13 precedes a period of 90% humidity, this observation is unexpected in light of the wealth of data indicating that high humidity suppresses programmed cell death (PCD) responses arising from either ETI or autoimmunity.

Our data do not contradict these previous results because we did not perform these experiments in constant high humidity, but rather under 12h 50%|12h 90% oscillating humidity. The readouts (i.e., pathogen growth and PCD) were measured after cycles of low and high humidity to capture the interplay between humidity oscillations and the clock, rather than as a direct effect of high humidity on ETI.

In addition, Xin et al. (ref 43) noted that at high humidity, AvrRpt2-associated ETI blocked the water-soaking symptoms caused by PstDC3000, suggesting that ETI would not be enhanced by increased local water availability.

Xin et al. indeed described that *Pst* AvrRpt2-mediated PCD is blocked under constant high humidity. However, it is important to note that moving these plants to 50% RH for 4 h was enough to trigger PCD (Fig. 3 b in that paper). In our experiment, plants were subjected to 12 h 50%|12 h 90% RH and PCD was quantified 36 h after infection (at the low inoculant used in this study) to capture the full effect of the interplay between LD and humidity cycles on immunity.

With regards to timing, the authors' observations also contradict previously documented circadian rhythms, such as the finding that genes associated with RPP4-mediated PCD exhibit peak expression at dawn in a CCA1-regulated manner, and that plants are more susceptible to Hpa infection when inoculated at dusk versus dawn (Wang et al., 2011; Nature 470:110-115). A reduced capacity to respond to *P. syringae* in the evening versus the morning has also been suggested (Bhardwaj et al., 2011 [PLoS One 6:e26968]; Zhang et al., 2013 [PLoS Pathogens 9:e1003370]). Greater clarity on the humidity/circadian relationship would be provided by bacterial growth assays that include plants maintained at a constant 90% RH. This would allow a comparison of the relative influence of high humidity versus the timing of inoculation on ETI.

We apologize for the confusion and have made changes in text to clarify (207-217). Experiments by Wang et al. (2011), Bhardwaj et al. (2012) and Zhang et al. (2013) were performed without oscillating humidity. Our data shows that the addition of high humidity at night, as occurs in nature, enhances ETI in evening-inoculated plants. Our experiments were not designed to compare levels of resistance in the morning vs in the evening, but rather to measure the contribution of oscillating humidity to immunity in morning- or evening-inoculated plants. This requires comparisons within the morning or evening samples which were exposed to either 70% or 50|90% oscillating RH (the same total amount of humidity) instead of comparing morning samples with evening samples. For the same reasons, we do not think it is necessary for our claim to include growth assays under 90% RH which have already been reported by Xin et al., 2016.

The choice of AvrRpt2 also may have influenced the results obtained from the ETI experiments. AvrRpt2 is known to modulate auxin signaling, and auxin responsiveness appears to be gated by the circadian clock (Covington and Harmer, 2007; PLoS Biol 5:e222). There is some suggestion that auxin can affect ETI (discussed in Naseem et al., 2015; J Exp Bot 66:4885:4896), although this remains to be conclusively demonstrated. Nonetheless, it would be worth repeating the growth assay and ion leakage experiments with a different ETI-eliciting effector to rule out any potentially confounding effects of other circadian-regulated components.

In the Cui et al., 2013 Plant Phys. paper discussed by Naseem et al., 2015, the *avrRpt2* effect on auxin was shown to be independent of the immune receptor RPS2. In our experiment, the effect on ETI by the oscillating humidity was abolished in the *rps2* mutant, indicating that the observed ETI phenotype in WT is unlikely due to this reported AvrRpt2 effect on auxin signaling.

To further address the reviewer's concern, we performed the ion leakage experiment using *Pf AvrRps4* in plants with and without the immune receptor RRS1 (TIR-NB-LRR) to show that humidity enhanced ETI in evening-inoculated plants is not specific to RPS2, which is a CC-NB-LRR, but rather a more general mechanism to strengthen immunity upon effector recognition (new Supplementary Fig. 9f).

Supplementary Figure 7: while the observation of type III induction by high humidity is interesting, it is not clear how this relates to the results shown in figure 6, in which all bacteria are infiltrated into the leaf apoplast as liquid suspensions (i.e. maximum humidity). Furthermore, if increased effector expression accounted for the differences in growth, then increased growth of Pst in Col-0 and Pst+AvrRpt2 in *rps2* mutants might be expected under natural humidity oscillations. This may be masked by the high levels of bacterial growth observed by day 3, in which case an earlier time point should be analyzed. At this point, the connection between humidity, effector expression levels, and ETI in the current manuscript appears to be overly simplistic, especially given the literature referenced above as well as the documented circadian fluctuations of other molecules such as salicylic acid and indicators of cellular redox state.

We cannot and do not compare the infection experiment shown in the old Fig. 6 (which was designed to test the effect of oscillating humidity on plant resistance) with the experiment in the old Supplemental Fig. 7 (which was performed to show the activation of the type III secretion by high humidity in bacteria). We use the latter as a possible explanation for the

increased ETI in evening-inoculated plants, which will need further investigation and is beyond the scope of this paper.

Materials and Methods Line 53: Figures 2-4, estimated period is calculated with FFT-NLLS. The standard approach with this tool is to apply a cutoff for the relative amplitude value produced by this analysis to eliminate arrhythmic luciferase traces. The authors need to indicate what they did for their analysis.

RAE cut-off was set to 0.7. We have added the cut-off to the legend of Figures 2-4 and Supplementary Figures 4-5. We have also updated the Method section with this detail (Line 338-340).

It is also not clear for each set of experiments what conditions the plants experienced prior to the time window shown in the figure. This seemed to change for each Figure, but was difficult to determine. The nature of the prior treatment can have a significant effect on the rhythms and clock resetting behavior.

The conditions that the plants experienced prior to the time shown in the window are described in the main text. We have updated figure legends for more clarity. Thank you.

Minor comments:

Line 55: change to "...circadian clocks alluded to effects of light, temperature, and air humidity on circadian clock regulation."

The reviewer's concern was addressed accordingly. Thank you for the suggestion.

Figure 1: Based on the figure legend, the y-axis of figure 1b should read: "Rosette weight (mg)" and figure 1c should read: "Rosette dry weight (mg)".

We have edited the y-axis of figure 1b and figure 1c as suggested by the reviewer.

Depending on the figure, error bars are standard deviation at times or standard error of the mean. A consistent method to represent uncertainty should be used throughout, preferably the more informative standard deviation.

Standard error of the mean was used in traces obtained from luciferase reporters as the focus was to show the mean trace. Standard deviation was used in all other cases.

line 416: "black" should read: "blue"

Corrected. Thank you!

line 452: the legend mentions a black arrow, but no arrows appear in figure 5. Please also describe the growth conditions prior to the application of the natural humidity oscillation.

Black arrows were added and the text edited accordingly. Thank you.

Line 461 and 480: abbreviation for standard error of the mean is missing the “m”.

We have edited the text and replaced s.e. with s.d. for more clarity.

Figure 6: Based on the figure legend, the y-axis of figure 6a should read: “Rosette dry weight (mg)”.

The reviewer’s concern was addressed and the y-axis changed to “Rosette dry weight”.

Supplementary Material, line 34: “CDD” should read: “CCD”

Corrected, thank you.

Supplementary Figure 7: Please indicate the genes whose fold change is statistically significant from one-fold. The error bars for hopD1 and HopN1 in HMM appear to overlap with the one-fold cutoff. Also, it would be helpful to know the fold-change of each gene for HMM versus KB media (at 50% RH). This would reveal the relative contribution of signals from the media versus humidity cues with regards to gene induction.

We addressed the reviewer’s concern by adding * to indicate fold change statistical significant from one-fold. To meet the reviewer’s interest in the fold-change of each gene for HMM versus KB (at 50% RH), we provide the figure below

Reviewer #3:

The paper presents a very interesting and worthwhile investigation of a potentially new input to the clock system. RH is also tested for its broader effects on Arabidopsis physiology and pathogen response, with promising results. The context from Harvard Forest met data is welcome. The studies are well-conceived, novel and original but raise significant technical issues that are not fully addressed. The present set of results are therefore more strongly interpreted in the text conclusions than seems warranted, and either the text or the results presented should be updated.

We appreciate the reviewer for recognizing the potential importance of our work and for finding our work well-conceived, novel and original. We agree with the reviewer’s concerns and we have performed additional experiments to address them as detailed below.

1. The mechanism and physiological importance of RH.
 - a. Leaf temperature clearly was warmer at 90% than 50% RH by about 1C (Supple Fig 2), as expected from greater transpirational cooling of the well-watered plants into the drier air at

50%. Stomatal dynamics in particular conditions will obviously affect this. If higher RH is associated with higher leaf temperature, this is sufficient to explain the sequence of gene expression peaks before/during/at the end of the high RH intervals in Figures 2 and 3. This would not be 'antiphase entrainment' as suggested in lines 124&ff, it is very similar to previous experimental results in temperature cycles (many references cited). It does result in clock entrainment in antiphase to the natural RH cycle, but the study was not conducted under natural conditions. This might also explain why Fig 4g-h show that RH entrainment continues in *cca1* mutants: temperature is known to affect many clock components, even if RH principally affected CCA1 expression.

We agree with the reviewer that the mechanism by which RH is sensed by plants is not yet known. Even with our additional temperature entrainment experiment (detailed below), which showed that RH is not sensed through the same mechanism as the ambient temperature, we cannot rule out transpiration as one of the possible RH-sensing mechanisms. Therefore, we have updated Discussion to reflect this point (lines 254-259).

b. The measured 1C change is small, but this measurement was in a separate study in a long interval at 50% or 90% RH. The temperature cycle during an RH cycle might be larger, at least transiently. Repeated, transient signals are clearly sufficient for entrainment in plants. Low amplitude temperature is also sufficient, as another example, Fujiwara, S. et al. Plant Cell 20, 2960-2971, doi:10.1105/tpc.108.061531 (2008) show that a 4C temperature cycle entrains sufficiently for well-coordinated rhythms even at the level of pooled RNA samples, a much less sensitive assay than the single-apex LUC measurements here or the seedlings tested in ref 27 Michael 2002. Smaller-amplitude temperature cycles are not known to be ineffective for entrainment in plants, simply no data are available: the authors could test this, see point 3a. A repeated 1C temperature cycle is sufficient to entrain mammalian cells (Saini et al. Genes Dev 2012).

As suggested by the reviewer, we performed the entrainment experiment using temperature cycles of $\Delta 1.5^{\circ}\text{C}$ (12 h 22.5°C | 12 h 21°C). We found that at the 1.5°C cycles, which is slightly higher than the 1°C temperature fluctuation observed during RH entrainment (Supplementary Fig. 2b,d,g), entrainment of *LHYp:LUC* in WT was weak (new Supplementary Fig. 6), suggesting that the observed entrainment to humidity cycles were unlikely due to temperature only. More importantly, we found that the 1.5°C temperature cycles could not entrain *LHYp:LUC* in *prp7 prp9* (new Supplementary Fig. 6), suggesting that humidity, rather than temperature, entrained the clock in our experiments.

c. The *prp7 prp9* control is worthwhile, as indeed it might have failed to entrain to RH. However, this mutant was reported to alter entrainment to 22C/12C cycles but not to 22C/28C cycles (refs cited). It is therefore unclear whether it is 'temperature-irresponsive at 21-22°C' as claimed. There is no clear interpretation of its continued entrainment to RH, though the data are clear.

Entrainment to 1.5°C temperature (12 h 22.5°C | 12 h 21°C) showed that the *prp7 prp9* mutant is temperature-irresponsive at 21-22.5°C (new Supplementary Fig. 6)

d. The response of CCA1:LUC in Fig 4 is far stronger than one would expect of a 1C temperature change, suggesting a specific RH input, but note the technical concern on the other genes below.

Also, there is no temperature record for any of the plants in these studies.

We recorded temperature, humidity, light, and soil moisture for every experiment. These data have now been deposited in the Biodare repository.

Conclusion. Points a and b above raise the reasonable possibility, or one could argue most parsimonious, testable hypothesis, that the mechanism for applied RH changes to affect the clock is via a small change in leaf temperature. That does not mean that the RH entrainment is not important physiologically. The authors might discuss which environmental conditions would give RH a significant entraining role alongside light and ambient temperature. Point d in contrast argues for a distinct RH effect, albeit with technical caveats.

Based on our new temperature entrainment experiment (new Supplementary Fig. 6), RH entrainment is unlikely to work through the same mechanism as the ambient temperature entrainment. However, as the reviewer pointed out, it is possible that the perceptions of RH and leaf temperature converge through transpiration. We have added this possibility to Discussion (lines 254-259).

Humidity is a signal that accumulates in antiphase from light and temperature with a couple of hours of overlaps at dawn and dusk, which may further distinguish these transition times from the rest of day and night, respectively (lines 241-246). With the addition of the humidity oscillation, we detected increased plant fitness and time-of-the-day regulation of effector-triggered immunity, indicating that the ability to sense oscillating humidity allows the clock to influence distinct physiological processes that are not controlled by the light/temperature-entrained clock (lines 246-250).

2. Physiological relevance

a. When RH is combined with LD cycles, the clock phase clearly follows LD alone, with no discernible RH effect on clock reporter phase in Fig. 5. The result would be the same for a very small temperature cycle.

Our new temperature experiment (new Supplementary Fig. 6) shows that humidity entrainment is distinct from temperature entrainment. Even though major emphasis has been put on studying clock period and phase, clock amplitude also has major physiological impacts (e.g., gating plant immune response; Zhou et al. Nature 523, 472). In this study, we show that reinforcement of the clock (Fig. 4, new Supplementary Fig. 8 on clock mRNA) by oscillating humidity significantly improves plant fitness (Fig.5), demonstrating the relevance of the humidity oscillation in LD.

b. Concluding that the RH cycle reinforces clock amplitude based on LUC alone is unsafe. The amplitude of LUC reporter rhythms is notoriously hard to interpret, as it depends on LUC-dependent factors (translation rate, and substrates O₂ and ATP) as well as the clock transcriptional regulators. The authors should note this possibility, as it will be harder to support their conclusion more directly. RNA assays in the simulated cycles might struggle to detect this change in amplitude. A transfer into constant conditions with some test of clock 'quality' (eg synchrony among LUC rhythms in individual plants) could be an alternative.

We agree with the reviewer and performed the suggested qRT-PCR. We found that humidity reinforced *CCA1/LHY* mRNA expression under simulated natural light-and-humidity cycles (new Supplementary Fig.8).

c. There is no control to test whether this effect is specific to the 'natural' phase relationship of LD and RH.

To show that the reinforcement of the clock is specific to the simulated natural light-and-humidity cycles, we added the data from the LUC experiment which was performed with the 12h 50%|12h 90% RH cycle in complete antiphase to the light cycle (new Fig 4d-f). We found that the 2-h overlap of light and high humidity is required for the reinforcement of the clock.

3. Technical concerns.

The discrepancies between clock markers in Figures 2 and 4 are hard to interpret or suggest variability somewhere in the assays.

a. The LL assay conditions are clearly effective, allowing for wide desynchronisation among individuals that persists after transfer to the imaging apparatus (as Fig 2 controls show). This gives the authors the opportunity to test whether applying a 1C (on average over 12h) cycle can entrain the clock.

Done as suggested. Please see our response to 1b above.

b. In tests of novel entraining stimuli, a common way to eliminate possible confounding effects is by demonstrating antiphase entrainment, in response to antiphase input cycles, and stable, opposite phases in subsequent constant conditions. Fig 3 goes in this direction, with the proviso that all genes would be expected to behave similarly in LL.

As mentioned by the reviewer, Fig.3 goes in the direction of demonstrating antiphase entrainment in response to antiphase input cycles. We believe that Fig. 3 along with Fig. 2 and Supplementary Fig. 4 address the reviewer's concern. Fig. 2a-c demonstrates stable and opposite phases when compared to reported phases in LD (~12 h difference between *CCA1/LHY* and *TOC1*, 12 h shift in the peak expression of *CCA1/LHY* and *TOC1*; respectively). Also, the stable and opposite expression of clock outputs in Fig. 2d-f suggests that all genes behave similarly. In Supplementary Fig. 4, we confirmed the opposite phase resulting from humidity entrainment using qRT-PCR.

c. If a changed RH cycle (Fig 3) or RH pulse (Fig 4) causes a phase shift in the clock that persists under constant conditions (as claimed in both figures), it is expected to alter all clock-controlled genes. Fig3 and Fig 4 show a phase shift in only two (Fig 3) or one (Fig 4) clock gene, or more specifically, in the transgenic lines carrying these reporters. This implies that the RH changes had a long-lasting effect on the coupling between the clock genes, such that some genes can change phase without affecting the others. During ongoing entrainment, such a change in waveform is well known. Under constant conditions, however, it is surprising and defies a simple biological explanation. Technical variability would be one possibility. It has been argued that such different behaviour among other reporters reflected control by distinct (often, anatomically separate) clocks. That seems unlikely for widely-expressed, core clock components. Lines 121-122 do not

address this concern.

We are not claiming that the phase angle observed between CCA1 and LHY after the humidity pulse is stable (long lasting) in free-running conditions because to make the claim a longer recording is required. Our data only support a correlation between the onset of high humidity with the peak expression of CCA1/LHY, suggesting an active induction of CCA1 /LHY by humidity. In Fig. 3, *TOC1p:LUC* was not shifted in the 16 h 50%|8 h 90% RH regimen (in contrast to *CCA1p:LUC/LHYp:LUC*) because the transition from high-to-low humidity was kept at the same time as in the day-neutral regimen.

d. Line 229. I doubt the SAM alone was imaged in a field of view comprising hundreds of plants. The apical region typically has high luminescence signal from the young emerging leaves surrounding the SAM, and these have stomata.

Whole rosettes (SAM plus leaves) were imaged. But we restricted the analysis to SAM by segmenting SAM alone using image J. However, we agree with the reviewer that we cannot completely rule out that the segmented image did not include young emerging leaves surrounding SAM. Thus, we have removed the paragraph about stomatal regulation from Discussion.

4. Interpretation of c-l-t- clock mutants, Fig. 2 and 6. E.g. Line 87 “these effects were clock-dependent”. These figures have valuable data but the specific phrase above is an overinterpretation, rather the effects depend on clock genes. Studies of a clock-less mutant (such as *cca1 lhy toc1*) cannot link any of the mutant’s phenotypes uniquely to a defect in circadian timing. Deficiency in these three TFs affects multiple, non-rhythmic genes, which could result in the loss of biomass/ responsiveness to changing humidity. Assays such as a T-cycle study of period mutants are required to show that the phenotype is specifically clock-related (rather than clock-gene-related), because a deficiency of timing can then be rescued predictably by a corresponding alteration in the environment. A larger panel of clock mutants under a single condition would be an intermediate step.

We rephrased to: “these effects were dependent on core clock genes” (lines 86).

5. Minor points.

a. The ‘natural humidity oscillation’ should probably be rephrased, for example, to ‘simulated natural conditions’.

We have updated the text accordingly.

b. The legend text and elsewhere should reference the FFT-NLLS method [Plautz et al. 1996], as this is materially different from what is described as a Fourier transform. The BioDare resource offers several phase-measurement options with FFT-NLLS, and the authors should specify which was used here.

We have updated the text to include the Plautz et al. 1997 citation. As described in the method section, we used the option of average peaks relative to time zero to report the calculated circadian phase.

c. I strongly support using the data repository to share data but users will retrieve a wide range of irrelevant, public data by logging in and performing the recommended search with “*”. It would be more direct if the authors listed the particular experiments that correspond to their figures, in the supplement. Those records can then be made publicly accessible, using the "Open Access publish" function, so no login is required.

Biodare experiments and corresponding figures are now matched in the supplementary files: “Mwimba et al. data”.

Reviewers' comments:

Reviewer #1 (Remarks to the Author):

The authors have for the most part addressed my concerns.

The response to the following point does not:

12. P9 lines 192-195 "However, it significantly enhanced ETI elicited by Pst carrying the AvrRpt2 effector (Pst AvrRpt2) in the evening-inoculated leaves (Fig. 6d), but not in morning-inoculated leaves (Supplementary Fig. 6c), suggesting a time-of-the-day sensitivity to ETI induction." Would be better to have these data in one figure, please. Similarly for Fig. 6e and Supplementary Fig 6d.

"We deliberately kept the results of morning-infected experiment in the old Supplementary Fig. 6 to avoid the implication that pathogen growth in the evening-inoculated and the morning-inoculated leaves could be compared. These experiments were not carried simultaneously and therefore are not comparable. The main point of this experiment was to compare defense under constant humidity (70%) versus oscillating humidity (50%|90%)."

The authors state that the experiments are not comparable, yet they do make comparisons between the results of them, so their response is somewhat confusing.

Minor:

Please supply the geographical co-ordinates of Harvard Forest to be supplied.

Reviewer #2 (Remarks to the Author):

In this revision, the authors have included some additional data to address concerns. They demonstrated that humidity-induced enhancement of ETI is not restricted to AvrRpt2 but is also seen with AvrRps4 (Sup Fig 9f). The TTSS gene expression graph in HMM versus KB is worthwhile to include in Sup Fig 9, so that readers do not get the mistaken impression that high humidity induces similar absolute levels of TTSS gene expression in both HMM and KB (since the levels are starting from a much lower point in KB at 50% RH).

Some issues still need to be addressed.

It is unclear why "It was expected that phases of CCA1p:LUC and LHYp:LUC were delayed with the transition from low-to-high humidity transition whereas the phase of TOC1p:LUC was unchanged." While this may be expected under driven conditions (as in Green et al. (2002) Plant Phys.), this is not necessarily so after transfer to free running conditions. The advancement of CCA1 and LHY phase occurring at the end of the final 90% period before free running conditions should lead to a similarly delayed phase for TOC1 after the removal of the 90% humidity treatment. TOC1 phase in LL does not show this in Fig 3c.

It is incorrect to argue that a PRC is not useful as "a direct demonstration per se that a cue is a zeitgeber", since the whole basis of this technique is to apply a treatment to change (i.e., set) the phase of the clock. For this manuscript, a PRC could give insight into the opposite phase effect of humidity. That being said, the humidity cycles given in these experiments do produce luciferase rhythms that are coordinated and achieve a stable phase. Secondary confirmation is important to prove the case – i.e. qPCR to validate luciferase reporters. It is concerning that the humidity pulse in Fig 3d, which elicits a substantial increase in CCA1p:LUC signal, causes a small phase change and this

treatment does not reset the phase close to that in panels a-c of Figs 2 and 3.

The qPCR shown in Sup Figure 4 is only for LHY and under much different conditions than all the other experiments in this manuscript (plants in LL for 2 weeks compared to LL for up to 96 hours), which is odd. Also, the expression amplitudes here are very small, making interpretation difficult. It seems better to confirm with qPCR what the LUC constructs appear to show for at least two (or more) transcripts under conditions identical to the figures. Sup Figure 7 seems to be a good case in point. Under conditions where Figure 3d shows strong induction of CCA1p:LUC, the qPCR result in Sup Figure 7a indicates little to no change in levels of CCA1 transcript.

It is fairly certain that the clock community would agree that the circadian clock is severely compromised, if not nonexistent, in a *cca1 lhy toc1* mutant. These plants may exhibit driven rhythms in LD but that is not the same as those produced by the internal clock. While the mutant plants are no better with oscillating humidity than without, these mutants do perform more poorly than WT overall. The point of the comment on Figure 6 was to draw attention to the possibility that if plants are sick enough it may not be reasonable to expect them to respond to any treatment.

Sup Figure 6 as presented does support the authors' conclusions that 1.5 degree temperature cycles weakly entrain LHYp:LUC. It is noteworthy that the phasing of the temperature entrained rhythms is comparable to that produced by humidity, when taking into account that the warm period in Sup Figure 2d coincides with the high humidity period. However, the data shown here are presented in a different manner than in Figures 2 and 3. The X-axes begin at ZT24, not -24. Also, the FFT-NLLS analysis was performed from ZT24-96 not ZT 0-72. This impairs direct comparisons with the other figures. Also, the highest amplitude rhythms are apparent at -24 to 0 and 0 to 24 in the other figures. In addition, eliminating the 0 to 24 time series will influence the outcome of FFT-NLLS analysis, both for period and phase analysis.

In response to a number of comments, the authors stated that their intention was to compare the effects of humidity oscillations versus constant humidity on plants inoculated in the morning or evening, emphasizing comparisons within time points rather than between time points. What is evident from the data, however, is that it is not just the humidity oscillation itself that affects immunity, it is the context of that oscillation that is important. Inoculations of plants in the morning before the shift from 90% to 50% RH do not seem to affect ETI, but inoculations in the evening before the shift back to 90% seem to enhance ETI. Given this difference, it is only logical to examine the significance of the humidity conditions in the period immediately following inoculation as a possible factor contributing to these observations, especially when humidity is well known to affect the outcome of plant-microbe interactions. In addition, data from Wang et al. (2011), Bhardwaj et al. (2012), and Zhang et al. (2013) were cited in the initial review as demonstrating that plants are more susceptible to infection in the evening versus the morning, and the authors dismissed these contradictory observations because they were obtained under conditions of constant humidity. It is important to note, though, that the infection experiments in the current manuscript also included light-dark cycles, and the humidity oscillations were said to reinforce the light-entrained circadian clock patterns. As such, it is not unreasonable to assume that these plants should behave similarly to those in previously published experiments, based on the idea that multiple zeitgebers can synergistically regulate the same circadian pattern. Yet, this is not seen, and instead there is the observation of enhanced ETI following evening inoculations. Regardless of the authors' insistence that comparisons of morning versus evening samples are not relevant to the discussion, discrepancies with the published data need to be addressed. (These complications could have been avoided if the conditions used for all of the clock gene expression experiments [constant light, oscillating humidity] had also been used for the infection experiments.) In consideration of all of these issues, I reiterate my recommendation to repeat the growth assay shown in Fig 5e/Sup 9c, with the inclusion of plants

grown at constant 90% RH. This would yield one of two informative outcomes: if the 50 to 90% RH shift is important, then constant 90% RH should abolish the enhanced ETI after evening inoculation; if high humidity does increase effector production, then enhanced ETI should be observed at constant 90% RH regardless of the time of inoculation. Although Xin et al. (2016) did do growth assays at 90%+ RH, they did not inoculate at different times of day nor did they perform growth assays with DC3000+AvrRpt2 under these conditions, so it would be worth performing. At the very least, there needs to be a detailed discussion of how the data in this manuscript can be reconciled with the existing literature.

On the topic of humidity and effector secretion, the authors state in the rebuttal:

"We cannot and do not compare the infection experiment shown in the old Fig. 6 (which was designed to test the effect of oscillating humidity on plant resistance) with the experiment in the old Supplemental Fig. 7 (which was performed to show the activation of the type III secretion by high humidity in bacteria). We use the latter as a possible explanation for the increased ETI in evening-inoculated plants, which will need further investigation and is beyond the scope of this paper."

As discussed above, the influence of humidity is highly relevant to experiments involving humidity oscillations. Moreover, any hypothesis to explain the results should be supported by all of the accumulated data. If humidity-induced effector secretion really does explain the increased ETI in evening inoculated plants, then Pst in Col-0 and Pst+AvrRpt2 in rps2 mutants should show enhanced growth under simulated humidity conditions versus 70% RH, but only when inoculated in the evening. This was not seen, and thus the humidity component should be acknowledged in the results as only a speculative contributing factor to the growth assay observations (it seems like a primary explanation in the current manuscript, lines 218-232).

Editorial comments:

line 297: should be "plant genotypes"

line 604: should be "amplitudes"

Sup Fig. 8 line 105: should be "plant samples"

Reviewer #3 (Remarks to the Author):

The authors have taken trouble to address many of the review comments. I do not think they have succeeded. Please see the Word review file attached for my detailed response and analysis.

[Editorial Note: Please see the rest of Reviewer #3's comments on the following page.]

The authors have not been able to address the substantive concerns about entrainment to the level required, despite additional experiments. They document substantial, physiological effects in growth and pathogen responses, and maybe those should take centre stage in a re-focused paper, with effects on the clock as one possible mechanism.

Major points

1. To address my earlier point 1, the authors compared rhythms after growth in LL, either under a programmed $\sim 1\text{C}$ temperature entrainment or programmed at constant temperature in new SFig6. This is a commendable control in response to my major concern. In contrast to the statement on line 144, the peak phase of pLHY:LUC was apparently set to the predicted fall in temperature (48h, 72h in SFig6). This, like their RH result, is what the authors term 'antiphase', and is the opposite of literature data on temperature entrainment. This internal contradiction is just confusing. One might suggest that the temperature input was irrelevant.

However, the luminescence data show objective evidence of entrainment by the 1C temperature cycle (see details below), in contrast to the authors' assertion, both in their analysis and when I analysed the timeseries data provided in the supplement. Taken together, the control experiment cannot be strongly interpreted for or against RH entrainment mediated by a change in leaf temperature. This equivocal result does not justify the strong assertion that Humidity is a novel zeitgeber.

2. The Discussion now includes my earlier point "It is possible that leaf transpiration may play a role due to changes in VPD. By affecting the leaf temperature through transpiration, humidity may also interact with light ...". I agree. Leaf temperature is not a novel zeitgeber. The title claim to novelty (or 'novel mechanism') is weakened.
3. To my original point 2. The authors' responses to the comments all reviewers made on apparently gene-specific effects, for example on clock phase in Fig. 3, have not addressed the main point. This is important because this Figure specifically addresses a change in clock phase due to RH entrainment, not clock consolidation/amplitude/etc, testing the claim that RH entrains the clock.

The difference in RH cycles can of course alter relative phases during entrainment (as in the Green 2002 reference cited in the rebuttal), and a pulse might affect only one gene. The data analysed in Fig 3 extended for 96h into constant light, however, and in Arabidopsis those phase changes rapidly relax in constant light as the system returns the light limit cycle (Edwards et al. MSB 2010; Dodd et al. New Phytol 2014). There is no evidence for the weeks-long 'aftereffects' that are observed in nocturnal rodents, for example. If the difference between RH cycles caused a stable phase change, it must affect the phase of all markers, but it doesn't in the data. On the second and later cycles, if LHY and CCA1 changed phase, so should TOC1. Progressive change in TOC1 phase over the data window would show up as a change of period, for example, even if the phase measure was unaffected, but a period change isn't shown either. I therefore agree with reviewer 2's several comments that entrainment (as opposed to consolidation) is not demonstrated, e.g. regarding old line 106.

4. The authors cannot argue that the LHY and TOC1 LUC reporters were too weakly expressed to detect a response to the RH pulse that was detectable in the RNA data, while still using these markers in many other figures. The new RNA data in S Fig 7 gives no indication the small change, in the 57h timepoint only, is statistically significant, nor that the effect on LUC

phase in SFig7b is significant. The RNA effect is gone by 60h even though the RH pulse continued to affect LUC in Fig 3d. These new results do not strengthen the paper.

5. There is no test of fitness here. The phrase should be removed from title and abstract, or modified to refer to the phenotypes tested as 'fitness correlates' or similar.
6. The new RNA data in SFig8 gives limited support as the effect size is small. The lme4 package does not provide p-values, so some detail is required on how the t-test was conducted.

Detailed points.

7. Arising from point 1, I realise that the authors must report the fraction of plants in each group that was interpreted as rhythmic in each analysis (or, for example, that unless noted otherwise, >80% of plants were rhythmic in all replicates). This is conventional where an RAE threshold has been used to exclude some of the period/phase values, as the authors do here, and for markers such as *Drosophila melanogaster* activity rhythms, where a minority of genotypically wild-type flies is expected to be phenotypically arrhythmic. It should reinforce the case for entrainment, as the plants grown in LL will likely have a lower rhythmic fraction.
8. The temperature cycle should not be described as 12 h 21°C| 12 h 22.5°C (line 123-4), as the data in SFig6 show a change of ca. 1.2C.

9. Supp Fig. 2

I think that the legend does not describe the study in Supp 2c-2e, as it seems that the RH programme changes at 60h, for two cycles. Also in S2d, do the temperature changes before this time perhaps reflect a 12:12 LD cycle?

Please show the leaf temperature scale in 2g from 20 to 24C, as in 2b and 2d.

While the range of leaf temperatures in S2g overlaps, they are significantly different (t-test $p = 0.017$). 6/10 leaf temperatures measured at 50%RH were lower than any temperature measured at 90%. The minimum at 50%RH is only 0.8C lower than at 90% (in a separate study, not during data analysis).

10. My original point 3d regarding the SAM.

This issue was not addressed sufficiently, e.g. in the new text "clock gene expression was measured by luciferase reporter activities in the shoot apical meristem (SAM), which has been shown to be the central timekeeping organ for plants" (Lines 106-108)." First, the rebuttal agrees that they image young leaves as well as the SAM. At most they might say "the apical region". Second, there is no evidence for a "central timekeeping organ" under entrained conditions, which are relevant in that context. Local entrainment overrides any systemic, coordinating signals in the rosette (Wenden et al PNAS 2012).

11. The authors should be commended for providing the data in the supplementary information. Based on that file, it seems they are using BioDare2 for analysis. This resource does not yet support public dissemination as original BioDare does, so the supplementary data file is key.
12. Analysis of data in Supp Figure 6, following my point 1.

As in my previous comment, a more powerful design is to test for the phase of the biological rhythm after treating groups of plants with opposite cycles of the candidate input signal, because this gives a clear expectation for a 12h phase difference between groups. In the authors' design, there's no clear

expectation of phase or period, rather the hypothesis is that rhythms will be consolidated (stronger, more uniform among individuals) after entrainment compared to no input.

I agree that the experiment in SFig6 does not show the strong entrainment by RH shown in Fig. 2. However, SFig6 seems to have weaker rhythms in all groups, compared to Fig 2b/SFig3 (there cannot be a strongly-entrained, positive control tested simultaneously, as there are only two matched chambers). I would therefore put limited credence on these results without replication.

The evidence for ambient temperature entrainment comes from the fraction of rhythmic plants. SFig6b, c show that more wild-type (LHY) plants were rhythmic after 1C entrainment (I estimate 22/24 plants were rhythmic after the temperature cycle, 14/22 without the cycle), and their phases seem more consolidated. The phase data look similar to main figure 2f. I repeated the analysis in the BioDare2 resource and obtained similar results using FFT-NLLS (right two columns below). In Zielinski et al. PLoSone 2011, we argue that none of the methods that we offer in BioDare/BioDare2 is ideal for rhythmicity detection of the type required here. We tentatively recommended the Lomb-Scargle periodogram, which shows 19/24 plants rhythmic after the temperature cycle, 10/22 without it. This indicates that prior temperature entrainment consolidated rhythms.

Plants	Temp cycle	Number in SFig6 (est.) from FFT-NLLS	Number tested (in .XLSX data)	Lomb Scargle periodogram	Rhythmic fraction	FFT-NLLS	Rhythmic fraction
WT	1C	22	24	19	19/24=0.79	22	22/24=0.92
WT	-	14	22	10	10/22=0.45	15	15/22=0.68
prr7;9	1C	10	20	10		12	
prr7;9	-	7	21	4		7	

The scatter plot above shows the larger number of plants with 'strong rhythms' of around 24h period with low RAE values in the blue series after temperature entrainment, compared to the brown series without entrainment.

Reviewer #1:

The authors have for the most part addressed my concerns.

The response to the following point does not:

12. P9 lines 192-195 "However, it significantly enhanced ETI elicited by Pst carrying the AvrRpt2 effector (Pst AvrRpt2) in the evening-inoculated leaves (Fig. 6d), but not in morning-inoculated leaves (Supplementary Fig. 6c), suggesting a time-of-the-day sensitivity to ETI induction." Would be better to have these data in one figure, please. Similarly for Fig. 6e and Supplementary Fig 6d.

"We deliberately kept the results of morning-infected experiment in the old Supplementary Fig. 6 to avoid the implication that pathogen growth in the evening-inoculated and the morning-inoculated leaves could be compared. These experiments were not carried simultaneously and therefore are not comparable. The main point of this experiment was to compare defense under constant humidity (70%) versus oscillating humidity (50%|90%)."

The authors state that the experiments are not comparable, yet they do make comparisons between the results of them, so their response is somewhat confusing.

We apologize for the confusion. The main point of the figures is to show that only in the evening-inoculated plants, we could see a significant difference in ETI between plants which were exposed to 70% and plants which were exposed to oscillating RH. Nevertheless, we agree to the reviewer's suggestion to put the two experiments side-by-side (see Fig. 6b,c; Supplementary Fig. 6b,c).

Minor:

Please supply the geographical co-ordinates of Harvard Forest to be supplied.

We have added the geographical co-ordinates of Harvard Forest in the text (see Lines 79-80).

Reviewer #2:

In this revision, the authors have included some additional data to address concerns. They demonstrated that humidity-induced enhancement of ETI is not restricted to AvrRpt2 but is also seen with AvrRps4 (Sup Fig 9f). The TTSS gene expression graph in HMM versus KB is worthwhile to include in Sup Fig 9, so that readers do not get the mistaken impression that high humidity induces similar absolute levels of TTSS gene expression in both HMM and KB (since the levels are starting from a much lower point in KB at 50% RH).

As suggested by the reviewer, we have included the additional data in the new Supp. Fig. 6a.

Some issues still need to be addressed.

It is unclear why “It was expected that phases of CCA1p:LUC and LHYp:LUC were delayed with the transition from low-to-high humidity transition whereas the phase of TOC1p:LUC was unchanged.” While this may be expected under driven conditions (as in Green et al. (2002) Plant Phys.), this is not necessarily so after transfer to free running conditions. The advancement of CCA1 and LHY phase occurring at the end of the final 90% period before free running conditions should lead to a similarly delayed phase for TOC1 after the removal of the 90% humidity treatment. TOC1 phase in LL does not show this in Fig 3c.

Our intent for the old Fig. 3 was to demonstrate that humidity could set the phase of the circadian clock. We have now more clearly demonstrated the resetting ability of humidity using the PRC experiment (new Fig. 2g,h) and the phase-resetting experiment (new Fig. 2e,f) suggested by this reviewer and reviewer 3.

It is incorrect to argue that a PRC is not useful as “a direct demonstration per se that a cue is a zeitgeber”, since the whole basis of this technique is to apply a treatment to change (i.e., set) the phase of the clock. For this manuscript, a PRC could give insight into the opposite phase effect of humidity. That being said, the humidity cycles given in these experiments do produce luciferase rhythms that are coordinated and achieve a stable phase. Secondary confirmation is important to prove the case – i.e. qPCR to validate luciferase reporters. It is concerning that the humidity pulse in Fig 3d, which elicits a substantial increase in CCA1p:LUC signal, causes a small phase change and this treatment does not reset the phase close to that in panels a-c of Figs 2 and 3.

We agree with the reviewer that a phase response curve (PRC) could give insight into the antiphase entrainment by humidity, thus supporting our claim that humidity is a zeitgeber. We have performed the experiment as suggested (new Fig. 2g,h). We also carried out the suggested qPCR experiment that supported the entrainment by humidity cycles in the new Supp. Fig. 3. In response to the reviewer’s other concern that the old Fig. 3d showed only a small phase shift, our explanation is that the plants was exposed to only a single 6-h perturbation by high humidity, which is in contrast to the 3-week humidity entrainment shown in the old Fig. 2a-c (new Fig. 2a-d) and Fig. 3a-c (now replaced by the phase resetting experiment proposed by reviewer #3; new Fig. 2e,f).

The qPCR shown in Sup Figure 4 is only for LHY and under much different conditions that all the other experiments in this manuscript (plants in LL for 2 weeks compared to LL for up to 96 hours), which is odd. Also, the expression amplitudes here are very small, making interpretation difficult. It seems better to confirm with qPCR what the LUC constructs appear to show for at least two (or more) transcripts under conditions identical to the figures. Sup Figure 7 seems to be a good case in point. Under conditions where Figure 3d shows strong induction of CCA1p:LUC, the qPCR result in Sup Figure 7a indicates little to no change in levels of CCA1 transcript.

As suggested by the reviewer, we have performed the qRT-PCR experiment in conditions matching Fig. 2 and found that mRNA rhythms support activity rhythms of the luciferase

reporters (see new Supp. Fig. 3). The experiment shown in old supp. Fig. 7a is now performed in DD in order to capture the induction of clock genes (*CCA1* and *TOC1*) by humidity (see new Fig. 2i). Because the clock quickly dampens in soil grown plants under DD (Zhou et. al., Nature 2015), we are able to detect the induction of clock genes more easily.

It is fairly certain that the clock community would agree that the circadian clock is severely compromised, if not nonexistent, in a *cca1 lhy toc1* mutant. These plants may exhibit driven rhythms in LD but that is not the same as those produced by the internal clock. While the mutant plants are no better with oscillating humidity than without, these mutants do perform more poorly than WT overall. The point of the comment on Figure 6 was to draw attention to the possibility that if plants are sick enough it may not be reasonable to expect them to respond to any treatment.

Niwa et al. Plant Cell Phys 2007 has reported that the triple mutant germinates well and grows normally on the MS agar as well as in soil. We have added this citation in the text (see line 201). Moreover, our data (Fig. 5a,d) show that the triple mutant has comparable rosette dry weight and seed mass to the WT under LD 70% RH. Thus, these plants are not significantly compromised in growth under our growth conditions to cause a significant concern. Because of its near WT morphology, the triple mutant is a great control for testing the contribution of the circadian clock on fitness-related traits in plants grown under LD.

Sup Figure 6 as presented does support the authors' conclusions that 1.5 degree temperature cycles weakly entrain LHYp:LUC. It is noteworthy that the phasing of the temperature entrained rhythms is comparable to that produced by humidity, when taking into account that the warm period in Sup Figure 2d coincides with the high humidity period. However, the data shown here are presented in a different manner than in Figures 2 and 3. The X-axes begin at ZT24, not -24. Also, the FFT-NLLS analysis was performed from ZT24-96 not ZT 0-72. This impairs direct comparisons with the other figures. Also, the highest amplitude rhythms are apparent at -24 to 0 and 0 to 24 in the other figures. In addition, eliminating the 0 to 24 time series will influence the outcome of FFT-NLLS analysis, both for period and phase analysis.

Thank you for pointing this out. We have re-run these experiments as suggested by the reviewer (new Fig. 3a,b). Imaging is now performed from 0 h to 96 h and analysis from 24 h to 96 h for both humidity and temperature entrainments. In all our analyses, we skipped the first cycle to eliminate the transient perturbation related to moving plants from growth chambers to the imaging chamber.

In response to a number of comments, the authors stated that their intention was to compare the effects of humidity oscillations versus constant humidity on plants inoculated in the morning or evening, emphasizing comparisons within time points rather than between time points. What is evident from the data, however, is that it is not just the humidity oscillation itself that affects immunity, it is the context of that oscillation that is important. Inoculations of plants in the morning before the shift from 90% to 50% RH do not seem to affect ETI, but inoculations in the evening before the shift back to 90% seem to enhance ETI. Given this

difference, it is only logical to examine the significance of the humidity conditions in the period immediately following inoculation as a possible factor contributing to these observations, especially when humidity is well known to affect the outcome of plant-microbe interactions.

We thank the reviewer for this great suggestion and we've examined the significance of the humidity conditions in the period immediately following inoculation. We found that the humidity oscillation post-inoculation, not high humidity by itself, is required for the enhanced ETI at night (new Fig. 6e, new Supp. Fig. 6d). Similar to the report by Xin *et al.* 2016, we found that high humidity suppresses ETI (Lines 248-250).

In addition, data from Wang *et al.* (2011), Bhardwaj *et al.*(2012), and Zhang *et al.* (2013) were cited in the initial review as demonstrating that plants are more susceptible to infection in the evening versus the morning, and the authors dismissed these contradictory observations because they were obtained under conditions of constant humidity. It is important to note, though, that the infection experiments in the current manuscript also included light-dark cycles, and the humidity oscillations were said to **reinforce** the light-entrained circadian clock patterns. As such, it is not unreasonable to assume that these plants should behave similarly to those in previously published experiments, based on the idea that multiple zeitgebers can synergistically regulate the same circadian pattern. Yet, this is not seen, and instead there is the observation of enhanced ETI following evening inoculations.

We have replaced “reinforce” with “increase in amplitude” (Lines 34, 65, 174, 177, 184, 191, 261, 285, 645) to more accurately describe our findings. This is important because the humidity-mediated increase in amplitude of core clock genes in LD does not necessarily imply that all outputs of the clock will be strengthened in the same way. Clock components can be activators and/or repressors of the same output genes and the level of these output gene expressions would depend on the interplay between the clock components influencing them. Thus, the effect of the increase in amplitude of a clock gene on a specific output needs to be tested experimentally. For example, increases in amplitudes of core clock genes triggered by SA lead to much higher induction of *PR1* in the subjective morning than in the subjective evening (Zhou *et al.*, 2015).

Results of our fitness-related traits support the synergy, which can be observed when the circadian clock is entrained by multiple zeitgebers (Yoshii *et al.* 2009). However, the expected synergy does not preclude possible specific effects of a zeitgeber. With evidence from the humidity regulation of ETI, we propose that the humidity oscillation regulates some outputs that are not regulated by light (Lines 36-38, 267-270).

Regardless of the authors' insistence that comparisons of morning versus evening samples are not relevant to the discussion, discrepancies with the published data need to be addressed. (These complications could have been avoided if the conditions used for all of the clock gene expression experiments [constant light, oscillating humidity] had also been used for the infection experiments.)

In consideration of all of these issues, I reiterate my recommendation to repeat the growth assay shown in Fig 5e/Sup 9c, with the inclusion of plants grown at constant 90% RH.

This would yield one of two informative outcomes: if the 50 to 90% RH shift is important, then constant 90% RH should abolish the enhanced ETI after evening inoculation; if high humidity does increase effector production, then enhanced ETI should be observed at constant 90% RH regardless of the time of inoculation. Although Xin et al. (2016) did do growth assays at 90%+ RH, they did not inoculate at different times of day nor did they perform growth assays with DC3000+AvrRpt2 under these conditions, so it would be worth performing. At the very least, there needs to be a detailed discussion of how the data in this manuscript can be reconciled with the existing literature.

As we mention above, we have performed the experiment suggested by the reviewer (new Fig. 6e and new Supp. Fig. 6d). Thank you.

On the topic of humidity and effector secretion, the authors state in the rebuttal:

“We cannot and do not compare the infection experiment shown in the old Fig. 6 (which was designed to test the effect of oscillating humidity on plant resistance) with the experiment in the old Supplemental Fig. 7 (which was performed to show the activation of the type III secretion by high humidity in bacteria). We use the latter as a possible explanation for the increased ETI in evening-inoculated plants, which will need further investigation and is beyond the scope of this paper.”

As discussed above, the influence of humidity is highly relevant to experiments involving humidity oscillations. Moreover, any hypothesis to explain the results should be supported by all of the accumulated data. If humidity-induced effector secretion really does explain the increased ETI in evening inoculated plants, then Pst in Col-0 and Pst+AvrRpt2 in rps2 mutants should show enhanced growth under simulated humidity conditions versus 70% RH, but only when inoculated in the evening. This was not seen, and thus the humidity component should be acknowledged in the results as only a speculative contributing factor to the growth assay observations (it seems like a primary explanation in the current manuscript, lines 218-232).

We agree that the humidity-mediated induction of effector secretion genes does not explain the increased ETI in the evening-inoculated plants. This is because our infection experiment was not setup to detect a direct effect that the environmental RH conditions have on the pathogen. In these experiments, the pathogen was pressure-infiltrated into the plant apoplast where humidity is expected to be at around 95%-100% RH (P.S. Nobel, W.H. Freeman and Company 1970) to by-pass stomatal regulation. Therefore, bacterial growth in these experiments was a measure of the overall plant immunity (minus stomatal defense), not a measure of bacterial virulence.

Editorial comments:

line 297: should be “plant genotypes”

line 604: should be “amplitudes”

Sup Fig. 8 line 105: should be “plant samples”

We have edited the manuscript as suggested by the reviewer

Reviewer 3:

The authors have not been able to address the substantive concerns about entrainment to the level required, despite additional experiments. They document substantial, physiological effects in growth and pathogen responses, and maybe those should take centre stage in a re-focused paper, with effects on the clock as one possible mechanism.

Major points

1. To address my earlier point 1, the authors compared rhythms after growth in LL, either under a programmed $\sim 1\text{C}$ temperature entrainment or programmed at constant temperature in new SFig6. This is a commendable control in response to my major concern. In contrast to the statement on line 144, the peak phase of pLHY:LUC was apparently set to the predicted fall in temperature (48h, 72h in SFig6). This, like their RH result, is what the authors term ‘antiphase’, and is the opposite of literature data on temperature entrainment. This internal contradiction is just confusing. One might suggest that the temperature input was irrelevant.

However, the luminescence data show objective evidence of entrainment by the 1C temperature cycle (see details below), in contrast to the authors’ assertion, both in their analysis and when I analysed the timeseries data provided in the supplement. Taken together, the control experiment cannot be strongly interpreted for or against RH entrainment mediated by a change in leaf temperature. This equivocal result does not justify the strong assertion that Humidity is a novel zeitgeber.

We are very sorry for the mistake made in labeling the time of the temperature records (shifted by 12 hr), which led to the reviewer’s misunderstanding of the result in the old Supp. Fig. 6. For clarity, we have moved the old Supp. Fig. 6c to the new Supp. Fig. 2b to show that we were able to control humidity and temperature in the Percival chambers. We have also re-adjusted its time 0 h for consistency with all our experiments.

Because of the oversight, we failed to communicate the main conclusion of the experiment, which is that the 1.5°C temperature change could entrain the circadian clock in WT but not in the *prp7 prp9* double mutant. Since humidity oscillation could entrain this temperature-insensitive double mutant, this result indicates that even if humidity entrainment is mediated by leaf temperature, it is not through the air temperature entrainment mechanism that requires PRR7 and PRR9 (Fig. 3a,b).

To rule out the involvement of leaf temperature as the entrainment signal for *prp7 prp9* during 50%|90% RH humidity oscillation, we repeated the humidity entrainment experiment

at 21°C, instead of 22°C, at which leaf temperature changes observed matched those under the 22.5|21°C 50% RH temperature entrainment experiment (new Fig. 3c,d). We found that this leaf temperature range could entrain only the WT but not *prp7 prp9*, while humidity oscillation, with the same leaf temperature range, was able to entrain the circadian clock in the *prp7 prp9* mutant (new Fig. 3b-e). This suggests that leaf temperature oscillation is unlikely the mechanism by which humidity entrains the clock (Lines 169-173, 276-284).

We now argue for the novelty of humidity as a zeitgeber not only based on the entrainment of *prp7 prp9* but also on the enhanced ETI at night, a phenotype distinct from the light and temperature entrained clock (Lines 270-273)

2. The Discussion now includes my earlier point “It is possible that leaf transpiration may play a role due to changes in VPD. By affecting the leaf temperature through transpiration, humidity may also interact with light ...”. I agree. Leaf temperature is not a novel zeitgeber. The title claim to novelty (or ‘novel mechanism’) is weakened.

The data presented in new Fig. 3 demonstrated to us that leaf temperature is unlikely to be the mechanism by which humidity entrains the clock and that humidity is a novel zeitgeber. As for leaf transpiration, we are still unable to determine whether humidity entrainment involves transpiration (Lines 282-284). We performed most experiments at constant temperature so that RH could be used as a proxy for VPD (Lines 73-75). Testing of the transpiration hypothesis is beyond the scope of this study.

3. To my original point 2. The authors’ responses to the comments all reviewers made on apparently gene-specific effects, for example on clock phase in Fig. 3, have not addressed the main point. This is important because this Figure specifically addresses a change in clock phase due to RH entrainment, not clock consolidation/amplitude/etc, testing the claim that RH entrains the clock.

The difference in RH cycles can of course alter relative phases during entrainment (as in the Green 2002 reference cited in the rebuttal), and a pulse might affect only one gene. The data analysed in Fig 3 extended for 96h into constant light, however, and in Arabidopsis those phase changes rapidly relax in constant light as the system returns the light limit cycle (Edwards et al. MSB 2010; Dodd et al. New Phytol 2014). There is no evidence for the weekslong ‘aftereffects’ that are observed in nocturnal rodents, for example. If the difference between RH cycles caused a stable phase change, it must affect the phase of all markers, but it doesn’t in the data. On the second and later cycles, if LHY and CCA1 changed phase, so should TOC1. Progressive change in TOC1 phase over the data window would show up as a change of period, for example, even if the phase measure was unaffected, but a period change isn’t shown either. I therefore agree with reviewer 2’s several comments that entrainment (as opposed to consolidation) is not demonstrated, e.g. regarding old line 106.

We have now performed both the PRC experiment and the phase resetting experiment suggested by the reviewer to explain the change in the clock phase due to humidity (new Fig. 2 e-h).

4. The authors cannot argue that the LHY and TOC1 LUC reporters were too weakly expressed to detect a response to the RH pulse that was detectable in the RNA data, while still using these markers in many other figures. The new RNA data in S Fig 7 gives no indication the small change, in the 57h timepoint only, is statistically significant, nor that the effect on LUC phase in SFig7b is significant. The RNA effect is gone by 60h even though the RH pulse continued to affect LUC in Fig 3d. These new results do not strengthen the paper.

We believe that there was a difference between responses to a single pulse and entrainment in our luciferase reporter experiments. While a single pulse was not sufficient to trigger a measurable phase shift in the expression of *LHYp:LUC*, the 10 days of exposure to humidity cycles were sufficient to shift the phase of *LHYp:LUC* by 12 h, coinciding with the same phase shift in *CCA1p:LUC* (new Fig. 2e,f). Therefore, we think it is justified to use the *LHYp:LUC* and the *TOC1p:LUC* lines in Figs 3 and 4. The mRNA data in Supp. Fig. 7 has been removed as suggested by the reviewer. Instead, we have performed the experiment in DD to show that a 2-h pulse of high humidity induces the expression of *CCA1* and *TOC1* (Fig. 2i). Because the clock quickly dampens in soil grown plants under DD (Zhou et. al., Nature 2015), we are able to detect the induction of clock genes more easily.

5. There is no test of fitness here. The phrase should be removed from title and abstract, or modified to refer to the phenotypes tested as ‘fitness correlates’ or similar.

To address the reviewer’s concern, we have replaced “fitness” with “fitness-related traits” throughout the manuscript.

6. The new RNA data in SFig8 gives limited support as the effect size is small. The lme4 package does not provide p-values, so some detail is required on how the t-test was conducted.

We performed *t*-test on means and s.d. obtained from the lme4 analysis (Line 423).

Detailed points.

7. Arising from point 1, I realise that the authors must report the fraction of plants in each group that was interpreted as rhythmic in each analysis (or, for example, that unless noted otherwise, >80% of plants were rhythmic in all replicates). This is conventional where an RAE threshold has been used to exclude some of the period/phase values, as the authors do here, and for markers such as *Drosophila melanogaster* activity rhythms, where a minority of genotypically wild-type flies is expected to be phenotypically arrhythmic. It should reinforce the case for entrainment, as the plants grown in LL will likely have a lower rhythmic fraction.

The rhythmic fraction is now reported in the new Supplementary Table 2.

8. The temperature cycle should not be described as 12 h 21°C | 12 h 22.5°C (line 123-4), as the data in SFig6 show a change of ca. 1.2C.

We have edited the manuscript to point out that the 12 h 21°C | 12 h 22.5°C covers the range of the temperature oscillation during humidity oscillation (Lines 153-157).

9. Supp Fig. 2

I think that the legend does not describe the study in Supp 2c-2e, as it seems that the RH programme changes at 60h, for two cycles. Also in S2d, do the temperature changes before this time perhaps reflect a 12:12 LD cycle? Please show the leaf temperature scale in 2g from 20 to 24C, as in 2b and 2d. While the range of leaf temperatures in S2g overlaps, they are significantly different (t-test $p = 0.017$). 6/10 leaf temperatures measured at 50%RH were lower than any temperature measured at 90%. The minimum at 50%RH is only 0.8C lower than at 90% (in a separate study, not during data analysis).

We apologize for this oversight on our part. We have edited the Supp. Fig. 2c as suggested by the reviewer. We also took advantage of the automation feature of a FLIR A320sc camera to obtain a denser and more reliable recording of leaf temperature (see new Fig. 3c,d) than our recording in old Supp. Fig. 2g obtained using the manually handled FLIR E90.

10. My original point 3d regarding the SAM.

This issue was not addressed sufficiently, e.g. in the new text “clock gene expression was measured by luciferase reporter activities in the shoot apical meristem (SAM), which has been shown to be the central timekeeping organ for plants” (Lines 106-108).” First, the rebuttal agrees that they image young leaves as well as the SAM. At most they might say “the apical region”. Second, there is no evidence for a “central timekeeping organ” under entrained conditions, which are relevant in that context. Local entrainment overrides any systemic, coordinating signals in the rosette (Wenden et al PNAS 2012).

We now refer to it as “the area around the shoot apical meristem, which displays robust and synchronous waveforms” (Lines 108-113, 358-360).

11. The authors should be commended for providing the data in the supplementary information.

Based on that file, it seems they are using BioDare2 for analysis. This resource does not yet support public dissemination as original BioDare does, so the supplementary data file is key.

Thank you!

12. Analysis of data in Supp Figure 6, following my point 1.

As in my previous comment, a more powerful design is to test for the phase of the biological rhythm after treating groups of plants with opposite cycles of the candidate input signal, because this gives a clear expectation for a 12h phase difference between groups. In the

authors' design, there's no clear expectation of phase or period, rather the hypothesis is that rhythms will be consolidated (stronger, more uniform among individuals) after entrainment compared to no input.

We thank the reviewer for the suggestion and have performed the phase resetting experiment as described by the reviewer (see new Fig. 2e,f).

I agree that the experiment in SFig6 does not show the strong entrainment by RH shown in Fig.2. However, SFig6 seems to have weaker rhythms in all groups, compared to Fig 2b/SFig3 (there cannot be a strongly-entrained, positive control tested simultaneously, as there are only two matched chambers). I would therefore put limited credence on these results without replication.

Experiments in Fig. 3a,b,e were repeated at least two times with similar results.

The evidence for ambient temperature entrainment comes from the fraction of rhythmic plants. SFig6b, c show that more wild-type (LHY) plants were rhythmic after 1C entrainment (I estimate 22/24 plants were rhythmic after the temperature cycle, 14/22 without the cycle), and their phases seem more consolidated. The phase data look similar to main figure 2f. I repeated the analysis in the BioDare2 resource and obtained similar results using FFT-NLLS (right two columns below). In Zielinski et al. PLoSone 2011, we argue that none of the methods that we offer in BioDare/BioDare2 is ideal for rhythmicity detection of the type required here. We tentatively recommended the Lomb-Scargle periodogram, which shows 19/24 plants rhythmic after the temperature cycle, 10/22 without it. This indicates that prior temperature entrainment consolidated rhythms.

We thank the reviewer for the analysis of our data. The result of the analysis also supports our conclusion that 1.5°C entrains WT whereas it fails to entrain *prr7 prr9*. As recommended by the reviewer, we have provided the Lomb-Scargle periodogram analysis in the new Supplementary Table 2.

Reviewers' comments:

Reviewer #3 (Remarks to the Author):

The authors should be commended on a keenly discriminating experiment in new Figure 3, although my interpretation still differs from theirs in two fundamental respects, and on clear presentation of results. In the initial submission, they argued that there was no significant change in leaf temperature under RH. Now they show clearly that 50/90 RH cycles alter leaf temperature (Fig. 3e; only prr79 is shown, I assume WT is similar) similarly to 22.5/21 air temperature cycles, and that these air temperature cycles entrain circadian phase in wild-type plants (Fig. 3b phase distribution; Sync Index in Supp Table 2 similar for temperature and RH) as the authors note in line 158. I recognise that effects of air temperature cycles might also be mediated by a mechanism independent of leaf temperature, such as an unknown mechanism driven directly by transpiration rate, but this is hardly a parsimonious explanation (see below).

The choice to plot 90%RH under LL as a predicted night (e.g. Figs 2, 3) refers to the weather data in nature. Because of this choice, the clock gene expression patterns are said to be "in antiphase" to LD entrainment (abstract, introduction, results etc).

However, Fig 4d-f shows there is no detectable effect of 90%RH in the night interval of a LD cycle. Fig 4a-c show increased amplitude in the 'simulated natural conditions' with 50%RH (cooler) dusk and 90%RH (warmer) dawn but no effect on phase. So the phase of the LD cycle, if this is present, dominates over the phase of the RH cycle. This is exactly the result for circadian phase that is expected from past work, where a strong LD cycle would be predicted to swamp the effect of a mild leaf temperature cycle. For amplitude, in contrast, the present results are novel and correlate with long-term physiological effects (Figure 5 in WT; my comments on the c-l-t- mutant from the first review, echoed by reviewer 2, still stand).

Parsimony suggests that leaf temperature is a sufficient explanation for the observations in wild type plants. The patterns of clock gene expression then follow expectation, e.g. with peak CCA1 and LHY at predicted dawn (= cool-warm leaf temperature transition = 50%-90% RH transition). It is not parsimonious to claim an unknown input pathway that results in counter-intuitive entrainment "in antiphase", if the results in wild type are sufficiently explained as normal-phase entrainment mediated by long-understood transpiration effects on leaf temperature and temperature entrainment of the clock. Rather, the authors document the molecular underpinning of the classical references to humidity and clocks by de Mairan and de Candolle (which I was not aware of).

The results in the wild type contradict the statement in the abstract "Interestingly, under constant light, air humidity oscillation shifts the phase of the clock to its antiphase independently of air or leaf temperature" (and similar elsewhere) in respect of both the antiphase and the independence from leaf temperature.

The novel result is that such a modest leaf temperature rhythm as is produced by physiologically-relevant RH changes could entrain the plant clock. Demonstrating that is a significant achievement, as I noted in my first review by comparison to similar data for mammalian cells. As I wrote in my first review, one might consider which natural conditions would make this most significant.

I agree that the prr79 mutant is insensitive to (or even disrupted by) air temperature cycles in the one experiment presented but sensitive to RH in the two experiments presented. If three temperature experiments were conducted with similar results (Fig 2 legend), they could be summarised in Supp Table 2. [an error in Supp Table 2, last line: prr79 rhythmic fraction $15/20=0.37$]. Nonetheless, the

results presented suggest either that there is a novel effect of RH that is revealed in the prr79 mutant due to its insensitivity to temperature change, or that RH has a larger effect on leaf temperature in prr79 than in wild type, so the mutant is more sensitive to RH despite its reduced temperature sensitivity. Because we're dealing with a mutant, there's inevitably a question about how much has been previously characterised.

Regarding the new PRC, I would not expect a subtle input stimulus to give a very reliable PRC. The obvious issue that all markers should show a similar shift is the same as raised in previous reviews, by me and reviewer 2. It's not clear if the phase change in the PRC is an advance or a delay, but a PRC with shifts in only one direction cannot mediate entrainment in general. Supp Fig 4 suggests it is a delay (plotted as a negative phase change by PRC convention), consistent with extended expression of CCA1. Entrainment of the 30h-period prr7prp9 plants to a 24h input requires a 6h phase advance in every cycle.

Contrary to the authors' assertion in the rebuttal, they did not conduct the experiment that I suggested in Figure 2e-f but a more laborious one with a weaker conclusion. I suggested using RH for the opposite entrainment (which they demonstrate instead with LD/DL), specifically because the two sets of plants would then have opposite phases. They show two sets of plants that have the same phase after re-entrainment by a single RH condition, after opposite LD/DL entrainment.

With some regret I feel that I cannot usefully contribute further to the publication process. I therefore withdraw from further review.

As an update to my earlier review, public dissemination has now been implemented in BioDare2, so data can be made Open on that platform.

Reviewer #4 (Remarks to the Author):

Mwimba et al have done exceptional work. They have made careful measurements. The key questions is whether the authors have demonstrated as claimed in the Introduction "Our study reveals that air humidity is a plant zeitgeber distinct from light and temperature"

The authors have clearly shown that humidity can affect the expression of clock genes. The current data would suggest that humidity does not entrain the oscillator and there is still a strong possibility that the effect of humidity are mediated by changes in leaf temperature. I reach this conclusion based on three factors

1. That humidity itself is not an entraining signal as demonstrated by the PRC.

In the abstract the authors state the following "A phase response curve for high humidity pulses shows resetting, indicating the presence of a humidity-sensitive circadian oscillator in plants." This is simply not the case based on the PRC shown in Figure 2g. This point was made clearly by Reviewer 3. First only CCA1:LUC showed a phase shift, but since the oscillator is a coupled network the phase shift by all components should be the same and LHY:LUC did not show a phase shift. This result is incompatible with humidity resetting the oscillator. Secondly there is confusion whether a 90% RH pulse causes an advance or delay of the circadian oscillator. In the PRC (Figure 2g) the authors report a 2h phase advance, which would push CCA1 expression closer to dawn. However, in other experiments (Figure 2) the authors argue that the 50% to 90% transition causes CCA1 to peak at dusk, and therefore it would be expected that 90% RH if it entrained the oscillator would cause a phase delay driving CCA1 expression towards dusk. Thus, if humidity were an entraining signal the finding in Figure 2g that 90% humidity causes a phase advance, is incompatible with the observation that in humidity cycles cause CCA1:LUC peaks at dusk. The most straight forward interpretation of these data is that humidity can affect CCA1 expression but does not contribute to entrainment. But

most concerning is the PRC is not reproducible, because in Supplemental Figure 4 the same experiment is performed in which the humidity pulse causes a phase delay. Completely conflicting responses in the phase response are not compelling data of entrainment.

The fact that the PRC is not reproducible and that the phase shifts are too modest to result in the antiphase behaviour the authors report suggests that humidity is not a novel zeitgeber. A further problem of the conclusion that humidity is a zeitgeber arises from the following statement in line "Under simulated natural light and humidity cycles, light sets the phase of the circadian clock whereas humidity increases the amplitude of the clock and further improves plant fitness-related traits." If a signal is a zeitgeber, it should be able to cause phase resetting to some extent in a light and dark cycle.

2. That the patterns of expression of the genes in response to humidity are not what would be expected if humidity is an entraining signal.

"Interestingly, under constant light, air humidity oscillation shifts the phase of the clock to its antiphase independently of air or leaf temperature, probably through the evening-induction of morning-phased circadian genes such as CIRCADIAN CLOCK ASSOCIATED 1."

This is not consistent with humidity having a physiological role in entrainment. Why would the oscillator have competing entrainment cues that result in antiphase to what would be required based on the environmental conditions? This strongly suggests that humidity can affect the expression of the clock components, but also strongly suggests it is not an entrainment mechanism.

3. Humidity affects leaf temperature and the expression pattern of the genes are suggestive of temperature entrainment.

Reviewer 3 is correct in pointing out that the expression pattern of the genes is strongly indicative of a temperature effect. Increased humidity increases leaf temperature through reduced evaporative cooling. Thus, it is essential to resolve whether the effects of humidity are a temperature effect. This is essential for interpretation of the data, because if it is a temperature effect, this completely rules out that humidity is a zeitgeber. However, even if humidity does regulate circadian clock genes, it does not demonstrate it is involved in entrainment. The authors have attempted to address this difficult to solve problem with a potentially very nice experiment, in which they investigate the effect of temperature in a *prp7prp9* double mutant, which cannot entrain to temperature signals. However, *prp7prp9* has more phenotypes than just temperature entrainment defects, and thus the experiment is not easy to interpret. I think this is probably one of the few approaches one could take at the current time and therefore there is merit in the experiment, but there are currently deficiencies in the data that must be addressed.

It is essential that the experiment is done in *prp7prp9* double mutants and in wild type with equal repetition and the experiment is such of critical importance to the interpretation of the data that at least three independent replicates are required of each experiment. Currently there is uneven replication of the experiment, one has been performed twice and one only once. I know it will drive the authors nuts that reviewers are asking for more experiments, but the authors make a fairly big claim, but the key experiment required to test this hypothesis has less than even the bare minimum repetition. Additionally, the single *prp9* and *prp7* null alleles are needed in this experiment, because they also have temperature independent phenotypes and that might affect the interpretation of the data.

I am also intrigued why an unusually high RAE of 0.7 was used to assess whether the signals were capable of generating a rhythm. In most labs > 0.5 is the cut off at which a rhythm is described as arrhythmic. It would appear that the authors are considering traces as rhythmic that in most labs would be rejected. Whilst there is no hard and fast rule, the authors could address this by using the standard in the field and providing all the data in a RAE v period plot as is standard in the field. This is the critical experiment that might suggest, though not demonstrate that the effects of humidity on the

oscillator are independent of temperature or not. To spell this out clearly the authors need at least three independent experiments for wild type and *prp7prp9*. It would also be necessary to include also single mutants, because *prp7* is an entrainment mutant but as far as I know has not been shown to affect entrainment by temperature. All the data should be presented showing the RAE for all the traces, either in a table or a plot.

Even if this experiment, does result in the same conclusion as at present from the limited data set currently, this would not show that humidity entrains the oscillator, it would show effects of humidity on the oscillator that might be temperature independent. Entrainment would require a PRC that has phase advances and delays and which is reproducible for different reporters of the circadian system. The data as presented in this manuscript are very interesting, and could be published if the authors did not claim a role for humidity in entrainment. The data do not support that conclusion for the reasons I have outlined. It would seem to this reader that humidity causes changes in *CCA1* expression, that may or not be a result of temperature-related effects. If one forces *CCA1* with forced cycles of e.g. humidity, this could set up rhythms in the oscillator that could persist, but it does not demonstrate that the signal is a zeitgeber that sets the phase of the circadian oscillator.

Comments of Reviewer #3

The authors should be commended on a keenly discriminating experiment in new Figure 3, although my interpretation still differs from theirs in two fundamental respects, and on clear presentation of results. In the initial submission, they argued that there was no significant change in leaf temperature under RH. Now they show clearly that 50/90 RH cycles alter leaf temperature (Fig. 3e; only *prp79* is shown, I assume WT is similar) similarly to 22.5/21 air temperature cycles, and that these air temperature cycles entrain circadian phase in wild-type plants (Fig. 3b phase distribution; Sync Index in Supp Table 2 similar for temperature and RH) as the authors note in line 158.

In the initial submission, we did not argue that there was no significant change in leaf temperature under RH. We actually showed $\Delta 1.5^{\circ}\text{C}$ leaf temperature changes during the 50%| 90% RH entrainment while maintaining the air temperature in the chamber within $\Delta 1^{\circ}\text{C}$. In the new submission, we submitted a new set of leaf temperature measurements with many more time points using an automated infrared camera. We originally thought that since the $\Delta 1^{\circ}\text{C}$ change of air temperature observed during the RH entrainment was well below the $\Delta 4^{\circ}\text{C}$ range used in previously reported temperature entrainment (Michael and McClung, 2002), air temperature was unlikely to be the signal explaining the RH effect. However, Reviewer 3 correctly pointed out that $\Delta 4^{\circ}\text{C}$ was not the experimentally tested minimal sensitive air temperature range; and changes in leaf temperature caused by humidity oscillation might mimic the entrainment by the air temperature.

Based on Reviewer 3's suggestion, we first subjected plants to $\Delta 1.5^{\circ}\text{C}$ air temperature and found it to entrain the WT, but not the temperature-insensitive *prp7 prp9* double mutant. Next, we tested whether the $\Delta 1.5^{\circ}\text{C}$ leaf temperature changes could entrain the clock using air temperature to modulate the leaf temperature. We found that the $\Delta 1.5^{\circ}\text{C}$ leaf temperature changes could entrain the clock in WT, but not in the *prp7 prp9* double mutant double mutant. However, humidity oscillation, which caused the same $\Delta 1.5^{\circ}\text{C}$ leaf temperature changes, could entrain the clock in *prp7 prp9*.

I recognise that effects of air temperature cycles might also be mediated by a mechanism independent of leaf temperature, such as an unknown mechanism driven directly by transpiration rate, but this is hardly a parsimonious explanation (see below).

The choice to plot 90%RH under LL as a predicted night (e.g. Figs 2, 3) refers to the weather data in nature. Because of this choice, the clock gene expression patterns are said to be “in antiphase” to LD entrainment (abstract, introduction, results etc).

However, Fig 4d-f shows there is no detectable effect of 90%RH in the night interval of an LD cycle. Fig 4a-c show increased amplitude in the ‘simulated natural conditions’ with 50%RH (cooler) dusk and 90%RH (warmer) dawn but no effect on phase. So the phase of the LD cycle, if this is present, dominates over the phase of the RH cycle. This is exactly the result for circadian phase that is expected from past work, where a strong LD cycle would be predicted to swamp the effect of a mild leaf temperature cycle. For amplitude, in contrast, the present results

are novel and correlate with long-term physiological effects (Figure 5 in WT; my comments on the c-l-t- mutant from the first review, echoed by reviewer 2, still stand).

Parsimony suggests that leaf temperature is a sufficient explanation for the observations in wild type plants. The patterns of clock gene expression then follow expectation, e.g. with peak CCA1 and LHY at predicted dawn (= cool-warm leaf temperature transition = 50%-90% RH transition).

It is not surprising to us that changes in air temperature that caused $\Delta 1.5^{\circ}\text{C}$ in leaf temperature entrained the WT circadian clock because temperature is a known zeitgeber. However, there is no evidence suggesting that this air temperature entrainment is through leaf temperature oscillation. In fact, whether leaf temperature can entrain the circadian clock has not been studied (to our knowledge). This was the reason that we used the temperature-insensitive *prp7 prp9* double mutant in this experiment. We found that *prp7 prp9* was not entrained by oscillating air temperature, but was entrained by oscillating humidity, even when both caused the same $\Delta 1.5^{\circ}\text{C}$ leaf temperature changes (now Supplementary Fig. 4). These data indicate that leaf temperature is not the signal for the RH entrainment in *prp7 prp9*.

It is not parsimonious to claim an unknown input pathway that results in counter-intuitive entrainment “in antiphase”, if the results in wild type are sufficiently explained as normal-phase entrainment mediated by long-understood transpiration effects on leaf temperature and temperature entrainment of the clock. Rather, the authors document the molecular underpinning of the classical references to humidity and clocks by de Mairan and de Candolle (which I was not aware of). The results in the wild type contradict the statement in the abstract “Interestingly, under constant light, air humidity oscillation shifts the phase of the clock to its antiphase independently of air or leaf temperature” (and similar elsewhere) in respect of both the antiphase and the independence from leaf temperature.

We based our statement on the genetic data obtained by comparing the WT control and the *prp7 prp9* mutant, which do not support the hypothesis that leaf temperature is the signal by which humidity entrains the plant circadian clock (see our detailed response above). However, we appreciate the reviewer’s concern and acknowledge the possibility of a pleiotropic effect in the mutant in the revised text.

The novel result is that such a modest leaf temperature rhythm as is produced by physiologically-relevant RH changes could entrain the plant clock. Demonstrating that is a significant achievement, as I noted in my first review by comparison to similar data for mammalian cells. As I wrote in my first review, one might consider which natural conditions would make this most significant.

As demonstrated in this manuscript, under natural conditions, humidity uniquely affects effector-triggered immunity at night.

I agree that the *prp79* mutant is insensitive to (or even disrupted by) air temperature cycles in the one experiment presented but sensitive to RH in the two experiments presented. If three temperature experiments were conducted with similar results (Fig 2 legend), they could be

summarised in Supp Table 2. [an error in Supp Table 2, last line: $\text{pr}79$ rhythmic fraction $15/20=0.37$] (authors: should be 0.75). Nonetheless, the results presented suggest either that there is a novel effect of RH that is revealed in the $\text{pr}79$ mutant due to its insensitivity to temperature change, or that RH has a larger effect on leaf temperature in $\text{pr}79$ than in wild type, so the mutant is more sensitive to RH despite its reduced temperature sensitivity. Because we're dealing with a mutant, there's inevitably a question about how much has been previously characterised.

The $\text{pr}7$ $\text{pr}9$ double mutant, which has been reported to be insensitive to temperature entrainment, was our only option to investigate genetically whether humidity signals through temperature. To take the reviewer's reservation about this mutant into consideration, we have toned down our claim that humidity entrainment is separate from the temperature entrainment (see Lines 147, 153-157, 166-169, 260-263, 270-274). Three temperature experiments were conducted and the data have been deposited in the Biodare repository. Results summarized in Supplementary Table 2 are for data shown in figures.

Regarding the new PRC, I would not expect a subtle input stimulus to give a very reliable PRC. The obvious issue that all markers should show a similar shift is the same as raised in previous reviews, by me and reviewer 2. It's not clear if the phase change in the PRC is an advance or a delay, but a PRC with shifts in only one direction cannot mediate entrainment in general. Supp Fig 4 suggests it is a delay (plotted as a negative phase change by PRC convention), consistent with extended expression of CCA1. Entrainment of the 30h-period $\text{pr}7\text{pr}9$ plants to a 24h input requires a 6h phase advance in every cycle.

In light of the reviewers 3 and 4's comments, we have decided to refrain from making the claim that humidity is a zeitgeber in the revised manuscript (see response to reviewer #4). As a result, we have removed the PRC experiment from the manuscript as well.

Contrary to the authors' assertion in the rebuttal, they did not conduct the experiment that I suggested in Figure 2e-f but a more laborious one with a weaker conclusion. I suggested using RH for the opposite entrainment (which they demonstrate instead with LD/DL), specifically because the two sets of plants would then have opposite phases. They show two sets of plants that have the same phase after re-entrainment by a single RH condition, after opposite LD/DL entrainment.

We disagree with Reviewer 3 that our experimental design produced a weaker conclusion. Our experiment, which started with the light entraining regimens (LD and DL) in two separate chambers before the humidity entrainment in a single one, allowed us to rule out any chamber-specific effects that plants might encounter if the experiment were conducted in the reversed order (i.e., humidity entrainment followed by LD and DL light entrainments). This experiment showed that humidity oscillation could set the phase of the circadian clock in LL (Fig. 2f).

As an update to my earlier review, public dissemination has now been implemented in BioDare2, so data can be made Open on that platform.

Thank you.

Comments of Reviewer #4

Mwimba et al have done exceptional work. They have made careful measurements. The key question is whether the authors have demonstrated as claimed in the Introduction “Our study reveals that air humidity is a plant zeitgeber distinct from light and temperature” The authors have clearly shown that humidity can affect the expression of clock genes. The current data would suggest that humidity does not entrain the oscillator and there is still a strong possibility that the effect of humidity are mediated by changes in leaf temperature. I reach this conclusion based on three factors

1. That humidity itself is not an entraining signal as demonstrated by the PRC. In the abstract the authors state the following “A phase response curve for high humidity pulses shows resetting, indicating the presence of a humidity-sensitive circadian oscillator in plants.” This is simply not the case based on the PRC shown in Figure 2g. This point was made clearly by Reviewer 3. First only *CCA1:LUC* showed a phase shift, but since the oscillator is a coupled network the phase shift by all components should be the same and *LHY:LUC* did not show a phase shift. This result is incompatible with humidity resetting the oscillator.

Indeed, a phase shift was only observed for *CCA1p:LUC*, but not *LHYp:LUC*, in response to a short pulse of high humidity treatment. We thought this data when combined with the observed phase shift of these two homologs, as well as other clock output genes, in response to 12 h 50%|12 h 90% RH cycles might be sufficient to claim phase resetting. The reviewer makes it clear that all clock genes have to show a phase shift in response to the short pulse of high humidity to claim phase resetting. Thus, we have removed the claim that humidity is a resetting signal of the plant circadian clock from our manuscript.

Secondly there is confusion whether a 90% RH pulse causes an advance or delay of the circadian oscillator. In the PRC (Figure 2g) the authors report a 2h phase advance, which would push *CCA1* expression closer to dawn. However, in other experiments (Figure 2) the authors argue that the 50% to 90% transition causes *CCA1* to peak at dusk, and therefore it would be expected that 90% RH if it entrained the oscillator would cause a phase delay driving *CCA1* expression towards dusk. Thus, if humidity were an entraining signal the finding in Figure 2g that 90% humidity causes a phase advance, is incompatible with the observation that in humidity cycles cause *CCA1:LUC* peaks at dusk. The most straight forward interpretation of these data is that humidity can affect *CCA1* expression but does not contribute to entrainment. But most concerning is the PRC is not reproducible, because in Supplemental Figure 4 the same experiment is performed in which the humidity pulse causes a phase delay. Completely conflicting responses in the phase response are not compelling data of entrainment.

The old Figure 2g, as well as the old Supplemental Figure 4, showed a phase delay. Phase shift was calculated as new phase (in response to humidity pulse) minus old phase (Old line 614 in old figure 2’s legend). Because we no longer make the claim that humidity is a zeitgeber able to reset the phase, we have removed the PRC from the main manuscript.

The fact that the PRC is not reproducible and that the phase shifts are too modest to result in the

antiphase behaviour the authors report suggests that humidity is not a novel zeitgeber. A further problem of the conclusion that humidity is a zeitgeber arises from the following statement in line “Under simulated natural light and humidity cycles, light sets the phase of the circadian clock whereas humidity increases the amplitude of the clock and further improves plant fitness-related traits.” If a signal is a zeitgeber, it should be able to cause phase resetting to some extent in a light and dark cycle.

We did not observe phase resetting by humidity under our current LD conditions. In future studies, we will test whether phase resetting by humidity is possible if lower light intensity is used. In the absence of these additional data, we have removed the claim that humidity is a natural zeitgeber from the manuscript.

2. That the patterns of expression of the genes in response to humidity are not what would be expected if humidity is an entraining signal. “Interestingly, under constant light, air humidity oscillation shifts the phase of the clock to its antiphase independently of air or leaf temperature, probably through the evening-induction of morning-phased circadian genes such as CIRCADIAN CLOCK ASSOCIATED 1.”

This is not consistent with humidity having a physiological role in entrainment. Why would the oscillator have competing entrainment cues that result in antiphase to what would be required based on the environmental conditions? This strongly suggests that humidity can affect the expression of the clock components, but also strongly suggests it is not an entrainment mechanism.

The primary role of humidity cycles in LD appears to be modulating the clock amplitude and improving fitness based on our experiments. We agree that we have not demonstrated phase resetting by humidity under LD. Thus, we have removed from the manuscript the claim that humidity is a zeitgeber.

3. Humidity affects leaf temperature and the expression pattern of the genes are suggestive of temperature entrainment.

Reviewer 3 is correct in pointing out that the expression pattern of the genes is strongly indicative of a temperature effect. Increased humidity increases leaf temperature through reduced evaporative cooling. Thus, it is essential to resolve whether the effects of humidity are a temperature effect. This is essential for interpretation of the data, because if it is a temperature effect, this completely rules out that humidity is a zeitgeber. However, even if humidity does regulate circadian clock genes, it does not demonstrate it is involved in entrainment. The authors have attempted to address this difficult to solve problem with a potentially very nice experiment, in which they investigate the effect of temperature in a *prr7prr9* double mutant, which cannot entrain to temperature signals. However, *prr7prr9* has more phenotypes than just temperature entrainment defects, and thus the experiment is not easy to interpret. I think this is probably one of the few approaches one could take at the current time and therefore there is merit in the experiment, but there are currently deficiencies in the data that must be addressed.

It is essential that the experiment is done in *prr7prr9* double mutants and in wild type with equal

repetition and the experiment is such of critical importance to the interpretation of the data that at least three independent replicates are required of each experiment. Currently there is uneven replication of the experiment, one has been performed twice and one only once. I know it will drive the authors nuts that reviewers are asking for more experiments, but the authors make a fairly big claim, but the key experiment required to test this hypothesis has less than even the bare minimum repetition. Additionally, the single *prp9* and *prp7* null alleles are needed in this experiment, because they also have temperature independent phenotypes and that might affect the interpretation of the data.

We thank the reviewer for acknowledging that testing the *prp7 prp9* double mutant is probably one of the few approaches one could take at the current time. With regard to the reproducibility of the experiments, the reviewer might have mis-read the legends. The humidity entrainments have been repeated five times, two times in Supplementary Fig. 4a and three times in Supplementary Fig. 4e; and the temperature entrainment experiments in the Supplementary Fig. 4b have been repeated three times. The data have been deposited in the Biodare repository, while results summarized in Supplementary Table 2 are for data shown in figures.

I am also intrigued why an unusually high RAE of 0.7 was used to assess whether the signals were capable of generating a rhythm. In most labs > 0.5 is the cut off at which a rhythm is described as arrhythmic. It would appear that the authors are considering traces as rhythmic that in most labs would be rejected.

$RAE \leq 0.7$ has been used to define the limit of rhythmicity in many studies (Mas et al 2003; Niinuma et al 2005; Xu X., Xie Q. and McClung C.R. 2010). Additionally, we show rhythmicity and synchronicity analysis performed on all plants in Supplementary Table 2. The $RAE \leq 0.7$ filter was only applied to plotted traces in Figure 2 and Supplementary Figure 4.

Whilst there is no hard and fast rule, the authors could address this by using the standard in the field and providing all the data in a RAE v period plot as is standard in the field. This is the critical experiment that might suggest, though not demonstrate that the effects of humidity on the oscillator are independent of temperature or not. To spell this out clearly the authors need at least three independent experiments for wild type and *prp7prp9*. It would also be necessary to include also single mutants, because *prp7* is an entrainment mutant but as far as I know has not been shown to affect entrainment by temperature. All the data should be presented showing the RAE for all the traces, either in a table or a plot. Even if this experiment, does result in the same conclusion as at present from the limited data set currently, this would not show that humidity entrains the oscillator, it would show effects of humidity on the oscillator that might be temperature independent. Entrainment would require a PRC that has phase advances and delays and which is reproducible for different reporters of the circadian system.

Please see our response above about the reproducibility of the experiments. We thank the reviewer for pointing out the limitation in using *prp7 prp9* to demonstrate that temperature and humidity perception are different. We have edited the manuscript to reflect this limitation.

The data as presented in this manuscript are very interesting, and could be published if the

authors did not claim a role for humidity in entrainment. The data do not support that conclusion for the reasons I have outlined. It would seem to this reader that humidity causes changes in CCA1 expression, that may or not be a result of temperature-related effects. If one forces CCA1 with forced cycles of e.g. humidity, this could set up rhythms in the oscillator that could persist, but it does not demonstrate that the signal is a zeitgeber that sets the phase of the circadian oscillator.

We have edited the manuscript in accordance with the reviewer's comment.